# A Formal Verification of a Reputation Multi-Factor Authentication Mechanism for Constrained Devices and Low-Power Wide-Area Network Using Temporal Logic

**DOI:** 10.3390/s23156933

**Published:** 2023-08-03

**Authors:** Wesley R. Bezerra, Jean E. Martina, Carlos B. Westphall

**Affiliations:** Campus Universitário—Trindade, UFSC—Federal University of Santa Catarina, Florianópolis 88040-380, SC, Brazil; jean.martina@ufsc.br (J.E.M.); carlosbwestphall@gmail.com (C.B.W.)

**Keywords:** multi-factor authentication, timed automata, formal verification, security

## Abstract

There are many security challenges in IoT, especially related to the authentication of restricted devices in long-distance and low-throughput networks. Problems such as impersonation, privacy issues, and excessive battery usage are some of the existing problems evaluated through the threat modeling of this work. A formal assessment of security solutions for their compliance in addressing such threats is desirable. Although several works address the verification of security protocols, verifying the security of components and their non-locking has been little explored. This work proposes to analyze the design-time security of the components of a multi-factor authentication mechanism with a reputation regarding security requirements that go beyond encryption or secrecy in data transmission. As a result, it was observed through temporal logic that the mechanism is deadlock-free and meets the requirements established in this work. Although it is not a work aimed at modeling the security mechanism, this document provides the necessary details for a better understanding of the mechanism and, consequently, the process of formal verification of its security properties.

## 1. Introduction

There is an inherent need for a lightweight approach to authenticating restricted devices in the Internet of Things (IoT). This is a research topic of growing importance today [1,2,3,4], stemming from different motivations, ranging from the fact that IoT devices are used as support for Distributed Denial of Service (DDoS) (Distributed Denial of Service is an attack where zombie devices are commonly used to send packets to a target, thus causing an unavailability of access to the target service [5,6,7,8]) to the lack of adequate security support for surveillance systems [9,10]. Thus, security in IoT systems presents challenges, especially for constrained devices (CD) and with connectivity through low-power wide-area networks (LPWAN).

Many advances have been suggested in the field of IoT. Examples such as the IoT softwarization [11]—a tendency that came from the network area—evidenced a need for updates of security strategies for IoT, as corroborated by the work of Kaur et al. [12] presenting challenges from different areas. It is still possible to cite the application of IoT in the solution of security problems, as demonstrated in the work of Gupta et al., which used watermarking and lightweight encryption to encrypt images.

However, IoT challenges must be addressed in different ways to resolve and guarantee the effective and proper functioning of the system, for example, through the formal validation of security proposals to be implemented. Thus, a formal assessment of the modeling of components and systems is of paramount importance, especially systems related to security [13]. Beyond the validation of the security protocol while verifying the system on a whole, evaluation can occur through several formal verification approaches.

This work aims to formally verify a multi-factor authentication mechanism with a Reputation (MFA_R) [14] despite the existence of many security protocol analysis programs, such as ProVerif (https://bplanche.gitlabpages.inria.fr/proverif/—accessed on 1 July 2023) an automatic security protocol verification tool that includes secrecy, strong secrecy, and authentication checks, among others. Through recent studies [15,16], it was noted that this tool is currently widely accepted in the research area, AVISPA (http://www.avispa-project.org/—accessed on 1 July 2023) a tool for formal validation of security protocols and systems that provides performance and scalability [17,18]. Its use in authentication [19,20] and in IoT [21,22] is significant, as it is in Tamarin Prover (http://tamarin-prover.github.io/—accessed on 1 July 2023) a verification tool where cryptographic protocols support first-order logic), and it has a growing number of publications in the area [23,24,25,26,27]. This study proposes to analyze aspects of the evolution of the system component states and their iterations, going beyond the information security aspects. This approach complements the analysis of protocols and fills the gap in the reliability and availability of security artifacts, which would be the MFA_R in this case.

As a contribution, the present study formally verifies the MFA_R mechanism. Other studies analyzed computational (processing time), spatial (memory usage), and network (packet analysis) aspects [14], and the security in message protocol [28,29] to be used in this mechanism—leaving such variables outside the scope of this study. This process was done through a detailed workflow of performing formal verification for security requirements on well-documented and replicable software components. Nevertheless, this study details the modeling that supported the construction of the formal model, its properties, and its queries. Correspondingly, the following contributions are more specifically presented:Threat modeling for MFA_R. This modeling supports the election of security aspects to be evaluated;Component modeling and description. In this modeling, internal aspects of the components are brought to light, as well as the functioning of the parts of the MFA_R, the exchange of messages, and the security domains;The behavioral modeling (state machines—STM) of the mechanism. Through this common language (UML2) to all areas of development, this study presents a view of the life cycle of components and details of operation of MFA_R, even for someone unfamiliar with modeling timed automata;The modeling of timed automata (TA). The construction of the automata model allowed the simulation through the chosen tool and the verification of the chosen properties through the threat-modeling process;Property verification via temporal logic (TCTL). This demonstrates the construction and conversion of security aspects required in threat modeling to TCTL properties, which can be verified automatically through the simulator and verifier.

The other sections of this document are presented as follows: Section 2 presents the materials and methods section, which includes related works, the components of the authentication mechanism, its state machines, and the timed automata modeled in Uppaal. Next, in Section 3, the performance of formal verification through TCTL is documented. Section 4 presents the discussion of the verification performed. Finally, Section 5 makes future work recommendations.

## 2. Materials and Methods

### 2.1. Background and Related Work

This section presents concepts and works that served as a basis for this study. Therefore, the content here addresses temporal logic along with its different approaches, and the threat model for MFA_R. However, we affirm that this study is not intended to exhaust the topics covered but rather to present the theoretical basis chosen by the authors and provide tools for readers who are not yet familiar with the topic.

In addition to the concept’s background, this work draws comparisons with related works. As these are separated by the approach utilizing temporal logic, each approach includes a short explanation of the concept and a set of related works that used the approach in question. Accordingly, for each type of approach, works are presented that may involve modeling, verification, simulation, or even algorithms.

All of the studies presented contribute to validating the assertion of temporal logic as a tool increasingly used to verify security requirements in software components and systems. Therefore, it is possible to verify that regardless of the approach, the relevance of formal verification is present in all works.

#### 2.1.1. Temporal Logic

To begin with, temporal logic is one of the ways of specifying modal logic that can be presented as linear time or branching time [30]. First, the linear time approach can be seen as computing a word formed from a formula to be evaluated; its processing can happen in the form of events or run-time. The second approach, branching time, creates branches that are not represented as a word but instead as a branching tree, consequently increasing resource consumption during the verification process. All the approaches discussed below are from these two branches of temporal logic spring.

Furthermore, the emergence of temporal logic occurred when the unified verification of concurrent and sequential programs was a great challenge [31] and the inadequacy of the current methods for verifying the real-time system was highlighted. Thus, different solutions for temporal logic were introduced over time, such as linear temporal logic (LTL), computational tree logic (CTL), timed computational tree logic (TCTL), and metric temporal logic (MTL), which will be presented together with some related works. As a result, for each logic approach, there is an introduction, a discussion of language aspects, and relevant related works; a table summarizing the works presented is at the end.

**Linear temporal logic (LTL)** is a type of logic that processes paths or words and was proposed by Pnueli [31] in 1977 in his prominent article “the temporal logic of programs”. As for its syntax, its connectives are *X* (next), which is unary, and *U* (until), which is binary. From these, one can derive Fφ (eventually) and Gφ (henceforth). *X* (next) checks if the next event/symbol of the word matches the one noted in the expression; *U* (until) checks if the event/symbol is maintained until a second symbol occurs; Fφ (eventually) checks if there is a future where the event/symbol is attainable; and Gφ (henceforth) checks if after reaching an event/symbol it is kept. Through these connectors, it is possible to describe properties and evaluate the dynamic aspects of the system.

Some works using LTL for security are noteworthy, such as the work by Grosu et al. [32], which presents an approach using Monte Carlo analysis through LTL formulas to identify attacks in the Needham–Schroeder authentication protocol. It is also worth mentioning Salva and Blot [33], who propose an approach using model learning and model checking to assess security through LTL in IoT systems. They evaluate three different systems against eleven security measures. Also, for Tun et al. [34] the use of LTL comes to express the desired behavior of the system, even at design time. In their study, the authors evaluate two systems for the weakening user obligations concept proposed in the work.

There is a study by Ouchani and Debbabi [35] on state-of-the-art analysis in the verification, quantification, and specification of software models in UML and SysML. The authors conclude that LTL is the most used approach for this purpose. Rounding out the LTL work, we have Kuze et al. [36] who presents a verification in runtime using LTL to detect false injection attacks in unmanned aerial vehicles (UAV). In these five studies, different ways of using LTL in security are presented, especially Grosu et al. [32], who, like this work, focus on authentication.

Following approaches in temporal logic, **computational tree logic (CTL)** is used to describe properties in branching time and was proposed by Emerson and Clarke [37] in 1981 and subsequently complemented in another work [38] by the same authors. Its notation was inspired by the unified branching-time (UB) [39] language. In this type of approach, each node of the tree presents a number of possible futures [40] that can have their properties expressed through the operators *E* (existential) and *A* (universal). For the existential operator, the property is valid for some branches being a possible result; from this branch, it is possible/probable to reach this symbol. As for universal, this property must be valid for all branches—it must always be satisfied. Also, operators like *X* (next), *U* (until), Fφ (eventually), and Gφ (henceforth) are also present in the grammar of this logic.

Several related works can be cited, such as the study by Maurya et al. [41], which models the security requirements using the CTL for the zero knowledge authentication protocol of Fiat–Shamir and verifies them through NuSMV. Regarding the following work, Mbongue et al. [42] validates a security architecture to prevent unauthorized access and data leakage in a multi-tenant cloud field-programmable gate array (FPGA) using CTL to verify the architecture and its configuration, in order to guarantee security in virtual machines (VMs). The authors Lilli et al. [43] make formal verification of the Z-wave protocol through the ASMETA framework, using CTL together with AsmetaSMV to verify security properties in the mentioned protocol.

Continuing this topic, Gava et al. [44] developed an algorithm called bulk-synchronous parallel (BSP) for on-the-fly computing, evaluating whether a structured security protocol model satisfies a CTL formula. This work addresses the issue of the cost of evaluating CTL formulas (in terms of time and space) using a parallelization approach. Finally, Valadares et al. [45] present a work that models Trusted Architectures for IoT using Petri Nets and describes behaviors (prohibited and desired) using CTL to evaluate key security points within this type of architecture. Most noteworthy are Maurya et al., who focus on zero knowledge authentication. Nevertheless, other works of the group present important contributions to the security and evaluation of resource consumption in CTL validations.

The next temporal logic is **timed computational tree logic (TCTL)**, which was proposed by Alur et al. [46] in 1993 and expresses time requirements using variables that change the time interval. Time is in the realm of reals with a differentiated form of quantification [30]. This change allows systems with time requirements to be evaluated using CTL and branching time. This approach introduces the possibility of operations with clock and time counting, allowing for the implementation of time constraints and the use of a time graph, and the evaluation of the problem is ascertainable within the established time constraints.

This type of logic associates real time with temporal logic, restricting the scope of the CTL operators in time, making it possible to evaluate the evolution of the systems and the clocks, and if the computation occurs within limits imposed by the time constraints. The syntax and usage of TCTL are similar to CTL, except for the aspect of time incorporated to support real-time, such as the incorporation of the concept of clock expressed by *x*. For example, AG(b→x.AF(c∧x<15)) expresses the query that verifies if, after reaching the *b* state, it will always reach *c* with a clock lower than 15.

Correspondingly, we can mention some studies, such as the one by AlQadheeb [47], which uses security policies in TCTL specific to users. In this work, the author proposes to improve security through formal modeling of user behavior and uses Uppaal for modeling. Also, Malik et al. [48] discuss modeling, through the Uppaal tool and property verification using TCTL, to verify the inter control center communication protocol (ICCP) (this protocol is used to exchange data such as control signals, data acquisition, and historical data). The work is presented in the form of a framework with the ability to be applied to other similar areas. Following the TCTL list, the work of Gu et al. [49] specifies the behavior of autonomous agents, and their safety requirements, for the context of autonomous vehicles. The specification is done through the Uppaal tool, and the properties are evaluated using TCTL. The verification of time requirements regarding time constraints in real-time systems was also an issue evaluated in this work.

Additionally, Park et al. [50] is presented, which exhibits a smart contract verification using timed automata with the Uppaal tool. This verification was performed for an auction based on smart contract ethereum using TCTL to evaluate the expected properties of this type of contract. Furthermore, Askapour et al. [51] proposed a tool that allows for the embedding of a complex robotic mission plan, based on formal methods, in a software engineering process. This work used Uppaal modeling and mission description through TCTL. Finally, there is the contribution of Camilli [52], who works in the context of process flow based on microservices. In his work, the author brings a framework for the continuous verification of microservices based on conductor (the engine open source of microservices orchestration—https://netflix.github.io/conductor/, accessed on 1 July 2023), using Petri nets for modeling and TCTL for verification. Among these six studies, Park et al. [50] address the verification of smart contracts using temporal logic and security.

In regards to **metric temporal logic (MTL)**, it is another approach that differs from the previous ones as it expresses temporal relations as causal [53]. The MTL was proposed by Koymans [54] in 1990 and complemented by Alur and Henzinger [55] in order to meet the need to express real-time properties in the model, being particularly important for this type of system [56]. Furthermore, this logic can represent both quantitative and qualitative temporal relationships [57] as well as future connectors and connectors representing past properties.

Another significant feature is its two possible semantics: pointwise and continuous −, much like TCTL. While pointwise semantics represent discrete time, continuous represents time in the rational domain. Its operators are similar to those contained in the LTL. However, they adopt modifications to represent the time interval and the past. The operators $◊\varphi_{\left[t1,t2\right]}$ (eventually) and $□\varphi_{\left[t1,t2\right]}$ (always) have the same interpretation, while the operators $⧫\varphi_{\left[t1,t2\right]}$ (eventually in the past) and $▪\varphi_{\left[t1,t2\right]}$ (always in the past) are used to represent the past. As per the mentioned operators, [t1,t2] represents the time interval specified in the expression. It is still worth mentioning that MTL uses the quantifiers ∀ and ∃ to describe its formulas.

Some important works regarding the security areas that use MTL are presented here, beginning with Ammar et al. [58] (which uses MTL to prove opacity linked to a defined Δ time while also providing proof of the complexity of the theme and a useful case in the context of cloud computing) and continuing with Ozmen et al. [53], which presents a service called IoTSEER (which models the behavior of IoT apps from their source code). The presented model is unified and allows for evaluating undesired states caused by physical interactions. The authors use MTL to express the policies evaluated in the Simulink (a modeling, simulation, and analysis tool—https://www.mathworks.com/products/simulink.html, accessed on 1 July 2023) from Matlab (*Software analysis numeric*—https://www.mathworks.com/products/matlab.html, accessed on 1 July 2023), which govern the physical iterations between apps helping in the deployment of the devices.

Another relevant work is that of Ahmed [56], which presents an intrusion detection system (IDS) solution based on temporal logic. His solution uses many-sorted first-order metric temporal logic (MSFOMTL) to represent attacks and temporal patterns concisely. Finally, Yahyazadeh et al. [59] present PatrIoT, a tool evaluating the actions of programmable IoT apps according to policies described in their language. Such language can be translated into MTL and is the basis for describing policies that deny executing actions that violate them. This type of logic highlights Ahmed’s work, which uses MSFOMTL for a security solution with IDS.

Among the tools utilized in the cited works, Uppaal is the one that stands out the most. Its utilization in works that adopt the TCTL is practically unanimous, characterizing it as the best choice for this type of logic. Other tools adopted in works that used the CTL, such as NuSMV, BSP, and ASMETA, should be mentioned. Simulink also appears in the adoption of MTL, and this simulation tool is commercially widespread in engineering projects.

Concerning the application, in Table 1 studies that have varying purposes within the security area were selected. The largest number of works is applied to the IoT area; however, areas range from network protocols, authentication, and security policies, to the security of standalone (software or hardware) agents. However, one can consider the emphasis on the formal verification of zero-knowledge authentication, which is in the same security sub-area of this work.

Although many works in the literature cover a range of verification and formal modeling aspects for security, such works focus on something other than authentication’s verification and modeling aspects in distributed IoT systems. Due to these limitations, our work is necessary and current to meet such aspects of this modeling.

Generally, the importance of formal security verification in different application areas can be noted. Additionally, it was found that the pair TCTL and Uppaal is an suitable choice for the formal verification of security in systems and hardware, according to several works listed above.

#### 2.1.2. Threat Model

This section describes the MFA_R threat-modeling process as the basis for creating the formally analyzed security requirements. Through a well-documented process, attack surfaces, threats, quantification through DREAD, classification with STRIDE, and prioritization with CVSS were considered. Accordingly, only after this further analysis was the security requirements for this project listed.

The analysis of the MFA_R vulnerabilities (Figure 1) took place in three steps: (1) the definition of the attack surface, (2) threat modeling, and (3) countermeasures. This approach was chosen for better comprehensiveness in the analysis of threats, and with this adoption, it now encompasses classification, severity assessment, and risk assessment. As a result, the priority of approach and the priority of resource utilization can be chosen in threats with greater impact. In this case, the evaluation and proof of mitigation of such threats is demonstrated in our authentication mechanism.

However, it is observed that in our modeling, the category of external actors (device) was not addressed. This scoping is focused on analyzing MFA_R threats only. Consequently, the category of external actors will not participate in this modeling. As a result, a threat model (TM) was obtained that focused more on the mechanism design and on some threats mitigated by design-time solutions, for which formal verification was performed. This analysis starts with the definition of attack surface.

The attack surface can be defined as potentially exploitable vulnerabilities in the solution’s artifact set. Thus, the definition of the attack surface (1) of this project can be seen in Table 2, which presents the correlation between the vulnerabilities and the analyzed attack surface. Even though such vulnerabilities can involve both hardware and software, this work is limited to the software components existing in the modeling and its relationship with data transmission technologies.

Additionally, in this attack surface modeling (Table 2) is grouped into three different categories: external actors, services, and data link. The category of external actors consists of the device that communicates with the MFA_R, which appears but is not detailed in the text. The category of services includes RegServ, AuthServ, and DataServ. Finally, the HTTPS and LPWAN protocols are in the link category. In general, this division into three parts is intended to be more didactic and understandable, bringing with it the ability to be reusable in future works that may extend or implement MFA_R.

As for the scope of the attack surface, it is important to note that the networking technology used in the LPWAN security domain was not specified. Several technologies bring embedded problem solutions as challenges to be overcome for the proper and secure authentication implementation. As some problems shown in the services were mitigated during the mechanism design, others may arise in this detail of the surface link. This fact will vary according to the technology used as some already have security measures implemented in their protocols while others do not.

The next item in the methodology is threat modeling (2), which is a tool that brings possible threats to the artifact under development at design time. Thus, although there are several threat-modeling methodologies, we chose to use three different methodologies, organized into three steps: **STRIDE (2.1)** classification, severity assessment with **CVSS (2.2)**, and risk analysis with **DREAD (2.3)**, and the result is detailed in Table 3 and Table 4, and is revealed in Table 5.

The first methodology is **STRIDE (2.1)**, which was used to categorize vulnerabilities contained in MFA_R and contains six different categories of threats [60]: spoofing, tampering, repudiation, information disclosure, denial of service, and elevation of privilege. Data flow diagrams (DFD) are used in its methodology to describe the system, its components, and the security domains through which the data travels. Comparatively, as it is a more consolidated approach, it has great acceptance by the community, many tools that support it, and many published articles about this methodology or that use it in their modeling.

The second methodology used was the common vulnerability score system, **CVSS (2.2)**, adopted for prioritizing the severity of the potential impact of vulnerabilities. This tool provides a systematic approach to assessing the severity of vulnerabilities using weights [61] ranging from zero to three (no severity and high severity, respectively) and is commonly used with other methodologies such as common vulnerabilities exposure (CVE) [62,63,64]. Therefore, due to not being a tool that classifies or proposes attack vectors, it is very commonly used in conjunction with other threat modeling methodologies (TMMs) during the threat assessment process in a project.

Third, the risk assessment **DREAD (2.3)** was used. This is an acronym for damage, reproducibility, exploitability, affect users, and discoverability [65,66], which are the categorizations used to quantify risk in this methodology. Furthermore, such a methodology can be viewed as a way of measuring the vulnerability risk associated with each associated threat [67] through the use of numerical values. Additionally, it is observed that each of the five categories is evaluated from 0 to 3, where zero is the absence of risk and three is the highest value (a more detailed quantification can be seen in the Table 4) since using this table helps reduce the subjectivity inherent to DREAD, which has been considered a drawback of this method.

In general, by analyzing the Table 5, it is noted that the greatest challenges are associated with the LPWAN(8) link attack surfaces and the data server(4), using CVSS prioritization as a basis. In both cases, data integrity is the central issue. However, individually, the threats that had the highest priority were “message tampering”(3) and “unauthorized access”(3). The first one (v09) can occur in any service that communicates through LPWAN in MFA_R, but it has greater relevance when publishing data in DataServ. The second one (v11) is a big problem at any point in the MFA_R workflow as its event invalidates its entire purpose. Subsequently, one can consider these two surfaces and threats as points of great interest in the design and evaluation of solutions in MFA_R.

In the last part, the countermeasures(3) are presented as the implementation of the security requirements arising from the threat analysis done previously. As a result, the main threats were chosen based on Table 5. Since these challenges were translated into security requirements for MFA_R, which resulted in the four requirements shown in Table 6, correspondingly, these aspects are key points of the mechanism. They attempt to literally translate the points where the mechanism cannot fail during the communication between all the components. Although translated into only four aspects, the requirements can map several vulnerabilities combined (Table 6—column 3) and guarantee that an eventual security breach is discovered.

**Table 4 sensors-23-06933-t004:** Risk quantification in DREAD [68]—table used as a basis for risk quantification in each category. The artifact is organized as follows: in the leftmost column is the list of categories, then the conditions classified as risk 3 (major), risk 2 (medium), and risk 1 (minor). If risk does not exist, it is classified as risk 0 (zero).

	(3)	(2)	(1)
D	full trust, admin level	leak sensitive information	leak trivial information
R	always reproducible, any time	reproducible in specific time and condition	difficult to reproduce
E	novice, in a short time	skilled, script	extreme skilled, no script
A	all users, confs, and key customers	some users, no default confs	few users, obscure feature, and only anonymous
D	published info explains attack, easy to spot	rare users, requires some knowledge	bug is obscure, unlikely potential for damage

**Table 5 sensors-23-06933-t005:** Vulnerability classification and prioritization. The classification according to STRIDE. Contrastingly, prioritization according to CVSS is presented, where priorities are assigned from 0 to 3. Finally, the quantification of threat impact risk through DREAD is presented. With this evaluation of three criteria, it is possible to have a complete understanding of the current threats in the MFA_R.

#	Surface	Vulnerabilities	STRIDE	CVSS	DREAD	Total Score
v01	s01	information disclosure	I	0	1.2	1
v02		leakage	I	0	1.4	
v03		DoS	D	1	1.8	
v04	s02	device impersonation	S	2	1.4	3
v05		brute force attack	E	1	2	
v06	s03	device impersonation	S	2	1.4	4
v07		tampering data	T	2	2.2	
v08	s04	information leakage	D	0	0.8	0
v09	s05	message tampering	T	3	1.6	8
v10		message fabrication	T	2	1.6	
v11		unauthorized access	E	3	2.2	

**Table 6 sensors-23-06933-t006:** Security aspects promoted by the MFA_R mechanism—the aspects represent key points in ensuring the security of the mechanism and combined cover a wide range of attacks.

#	Aspects	Threats
a01	all data publishing must be done only by authenticated devices	v03, v04, v07, v09, v11
a02	credential generation should be identified and mitigated	v04, v05, v06, v11
a03	credential theft should be identified	v04, v06, v11
a04	publication of random data should be identified and mitigated	v10
—	out of scope	v01, v02, v08

In short, methodical threat analysis was carried out to support formal verification. As such, it was possible to approach from the definition of the attack surface, going through the threat analysis process in three stages, ending with the countermeasures expressed through the aspects table (security requirements). Moreover, although being a long process, the steps were presented in an intelligible, step-by-step, well-documented way so they could be reused in future works.

### 2.2. Methodology—Development Process

This section describes the steps and tools used during the experimentation (as seen in Figure 2). The methodology is simple, aiming at validating the hypothesis of adequate functioning and guaranteeing the following security aspects described in Table 6.

In general, this study focuses on the dynamic and behavioral aspects of modeling the mechanism, leaving some static details outside its scope. The modeling workflow took place in four steps: (a) the modeling of the mechanism components; (b) abstract modeling with a state diagram of the mechanism’s functioning; (c) modeling with timed automata; and (d) formal verification with temporal logic. In this manner, it was possible to trace the evolution of the modeling from the components and their messages to the formal verification. It is also observed that if any specialist wants to develop verification through another formal technique, it is possible to use the modeling from the STMs without any problems. This issue happens because the models are not dependent on any technology.

Initially, the parts of the mechanism are presented through a **component diagram (a)**. This diagram expresses the communication between the software artifacts and where the components were deployed in each device (physical or virtual). Such information was used as an input to build the threat model and the other diagrams that follow.

For **state modeling (b)**, the UML2 language was used. Specifically, the diagram of state machines was used, a behavioral diagram with which it is possible to analyze the life cycle of each component. Since these different software components are on different devices, they present great potential for interlocking. Due to this potential, this abstract overview becomes fundamental to error-free modeling.

Employing **timed automata (c)**, the behavior and communication of each component expressed temporal characteristics. The state machines described in the previous step served as inputs to the TA construction process and underwent adaptations to alter the process’ modeling in automata. Some aspects of STMs were suppressed in creating models that supported queries in temporal logic. It was also possible to implement guards (restrictions for the evolution of an automaton) and the simulation of the operation through the tool built into Uppaal (https://uppaal.org/, accessed on 1 July 2023).

Finally, there is the **property verification (d)**, which happened using temporal logic. The properties checked were derived from the security aspects mentioned above (Table 6). The queries were described in a TCTL reduction implemented in Uppaal and executed in the verification environment of this same tool.

Upon the completion of the four steps, it was possible to present the entire proposed verification cycle. Additionally, it was also possible to describe each component and its behavior, convert behaviors into automata, and then verify the listed safety aspects.

### 2.3. MFA_R Processes

This subsection comments on the MFA_R processes. These processes are divided into four parts: registration process, authentication process, enforcement process, and data publication process. As authentication takes place through multi-factors using reputation, a greater detail of this process will be done in the following part of the text.

In general, each process has its algorithm or set of algorithms commented and detailed. However, each algorithm does not stick to the construction details of a programming language but serves as a guide for implementation in any language of the structured paradigm or its subset. While using necessary adaptations, it is possible to apply the MFA_R to any programming paradigm and any language as the abstraction level of the algorithms supports this. The set of symbols and terms used for the algorithms can be consulted in Table 7.

#### 2.3.1. Registration Process

Registration is the first step in using the proposal and can be performed when unpacking to use the device. It must occur through a network with greater traffic capacity or even through a secure channel (using SSL), reducing the chance of copying the transferred registration data.

This is a process that demands the transmission of a larger number of data; therefore, it is recommended not to use a network charged by packets or with daily data traffic quotas [69,70], which can significantly increase costs when activating multiple devices in a batch.

Three pieces of equipment and two different devices are required for the registration process. The two devices are the device to be registered ($dev_{\mathrm{reg}}$) and the gateway (gw). The three components are the device client firmware ($dev_{\mathrm{reg}}$), the registration server (regServ), and the gateway database ($db_{\mathrm{gw}}$). Note: the registry service does not need to be located in gw. However, in our proposal, we use this architecture to reduce implementation costs in real situations.

The process starts after connection to the registration service (regServ) by the unregistered client ($dev_{\mathrm{unreg}}$). O $dev_{\mathrm{unreg}}$ sends a message (there are several formats used to exchange messages, and in our study, we used JSON, due to the low overhead generated) for the regServ of gw. Upon receiving the message, regServ consults its database ($db_{\mathrm{gw}}$) to check if the device has already been previously registered. If so, the registration is denied and a message with −1 is returned to $dev_{\mathrm{unreg}}$. This is done to avoid device cloning and threats related to this aspect.

The query is executed by generating a hash of the sent data. If it is already registered in the database, this means that the device in question has already been registered. This solution does not have a recommendation for device re-enrollment situations, this being a procedure to be resolved for each situation in the future. If $dev_{\mathrm{unreg}}$ has not been previously registered, a random password and username based on its hash is generated. Data encapsulated in a record object, along with data coming from the device Equation (Equation 1), is sent to the database ($db_{\mathrm{gw}}$).
(1)obj_reg<user,passwd,reg_data>

At the end, the timestamp, user, and password are returned to $dev_{\mathrm{unreg}}$. These data are encapsulated in a registration confirmation message Equation (Equation 2), and, from that moment on, the device is considered registered ($dev_{\mathrm{reg}}$).
(2)reg_confirm<timestamp,user,passwd>

This is a process that should, at most, happen once. With devices in compatibility mode, registration must be carried out through a user interface. The service that will provide the registration capability is the same; however, a web layer must be built to allow the IoT system administrator to register participating legacy devices. As such, regarding the service, the difference should not be noticed, and if the registration is carried out by a web layer or a RESTful API, this should be transparent for the service.

For a better understanding, the two registration algorithms are presented here (Algorithms 1 and 2), which are client-side (firmware) and server-side (server), respectively. Through this pseudocode, it is possible to notice some implementation details that are more easily presented in the form of code (i.e., function callbacks).
**Algorithm 1:** Device registration process—client-side
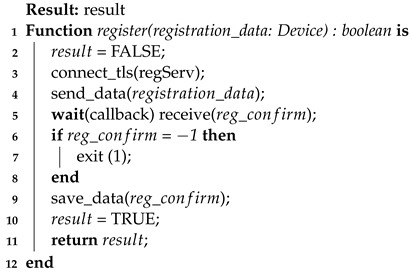


In the first presented Algorithm 1, the client-side registration produces a simple flow of this process. However, it is important to highlight some details in the code, such as the secure connection (line 3) and the saving of record data at the end of the process (line 9). These are details that reinforce the need to use a secure channel at the time of registration as well as the need to save registration confirmation data (Section Equation 2) on the client side as well.

The second Algorithm 2 presents the process from the server’s point of view (gw). This algorithm explains the processes of verifying the existence of a previous record (already registered—line 5), generating the record data (lines 8 to 10), and saving the data in a database on the server side (line 11). In this manner, it was possible to formalize the necessary steps on the server for the registration process.

The registration process is the first step of all authentication and where data between client and server are exchanged. It is of paramount importance to fully understand this process, as well as the functions and components it will require (i.e., hash functions and data structure) for its proper functioning. Through this understanding, we will be able to better understand the details of the other authentication processes (Section 2.3.2), enforcement (Section 2.3.3), and data publication (Section 2.3.4).
**Algorithm 2:** Device registration process—server-side
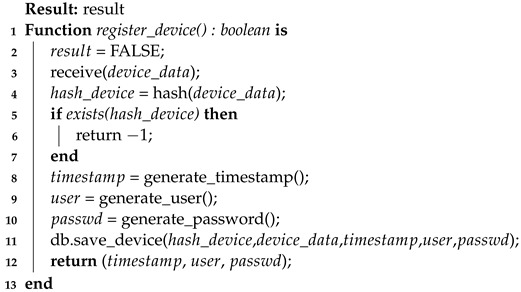


#### 2.3.2. Authentication Process

Authentication starts with a registered device ($dev_{\mathrm{reg}}$) sending authentication data to the related service (authServ). Three distinct flows can occur from that moment on: (i) failed authentication; (ii) successful authentication—compatibility mode; and (iii) successful authentication—reputation mode.

When authentication fails (i), authServ should store an audit log (this log should be in a textual format to be processed by an IDS, which works together with the authentication mechanism to prevent DDoS, among other attacks). Upon successful authentication in compatibility mode (ii), authServ returns a message confirming the authentication and communicates with the authentication inspector (authInspector) and records the auth_code (the authentication code is stored in the authInspector software component, and this code acts as the device’s authenticated) device credential in compatibility mode.

authServ generates an authetication_query for $dev_{\mathrm{reg}}$ from the data exchanged at the time of registration. Upon receiving it, $dev_{\mathrm{reg}}$ must generate a response to the query and send it to authServ, which will then compare it with the expected result, previously calculated. If the values are equal, authInspector is informed that $dev_{\mathrm{reg}}$ has risen to the AUTH2 level. It is important to highlight that when accessing AUTH2, the device becomes eligible for AUTH3, according to its behavior when sending data. In the case of a negative result during authentication, −1 is returned as an error code (we note that the value −1 as the authentication error pattern prevents the identification of the access level that the device has because the errors are the same for the different levels; someone listening to the network will not be able to obtain the access level information based on the capture of the failed returns).

During the publication of data, the **third factor** is validated. After reaching the AUTH2 level, the sensor receives a reputation score. As posts are made and the data stay within the standard deviation, $dev_{\mathrm{reg}}$ gains more reputation, and when it reaches the required upper threshold (Equation (Equation 3)) it becomes eligible for the AUTH3 level.
(3)upper_threshold≤score−salt

There is also the possibility of a device going down from the AUTH3 level to the AUTH2 level, which will occur if the sensor fails and sends anomalous measurements, causing the sensor to lose reputation. If this is recurrent, the sensor may reach the lower threshold (Equation (Equation 4)) and drop to the AUTH2 level. Such level changes should generate an audit log.
(4)lower_threshold≥score+salt

In this way, the general aspects of the authentication process in its three stages are contemplated. Also, the thresholds and their functioning were brought together with the reputation score during the device authentication process. For further clarification and formalization of the process, three algorithms are presented below to clarify the authentication process: client-side authentication (Algorithm 3), the first step of server-side authentication (Algorithm 4), and the second step of server-side authentication (Algorithm 5).
**Algorithm 3:** Device authentication process—client-side
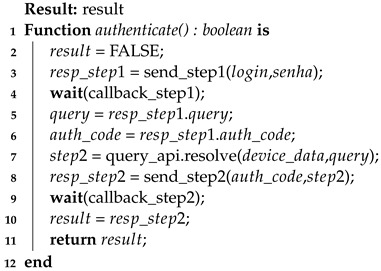


**Algorithm 4:** Device authentication process (STEP1)—server-side

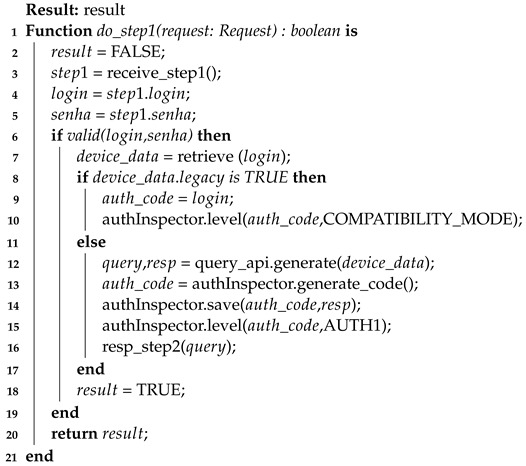



    The algorithm that describes the behavior of the device during authentication (Algorithm 3) has important aspects that must be observed. Some of these aspects involve the callback functions that must be handled (lines 4 and 9) and the use of the query API to resolve the second authentication factor (line 7).

The second algorithm (Algorithm 4) provides password-based authentication and ensures compatibility with legacy devices. As can be seen, after validating the device login (line 6), the server checks if the device is in compatibility mode (line 8). If it is a legacy device, the device will be registered in authInspector as compatibility mode (COMPATIBILITY_MODE). If it is not legacy, a challenge–response factor will be generated through the query API (line 12), saved in authInspector along with the device access code (line 14), where it is still registered that the device is in the first level authentication (line 15). In the end, the query is sent to the device to solve it (line 16).

Finally, there is the resolution algorithm for the second authentication factor (Algorithm 5). This second factor is analyzed through a single algorithm, the server-side, which shows the steps taken by the server in validating the challenge–response (CR) factor. After receiving the response from the device and decoding it (lines 3 to 5), authInspector is consulted to validate the response using auth_code as a search key. Once validated, a record is generated that the device is authenticated in the second factor (line 7) and an initial reputation score is assigned to it (line 8). Note that the interaction between authServ and authInspector is essential for the success of the authentication process and the maintenance of the reputation score, as shown by the previous algorithms.
**Algorithm 5:** Device authentication process (STEP2)—server-side
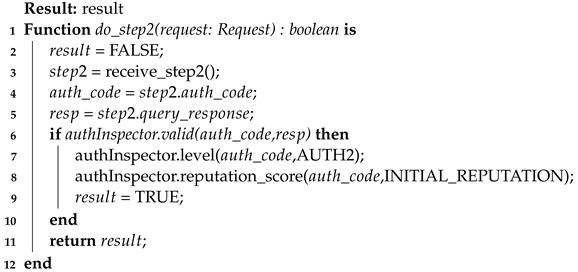


#### 2.3.3. Enforcement Process

Even if the authentication is successful, a procedure is needed to reinforce the system’s security regarding device identification. This process is carried out after a specific number of messages or a specific time of authentication of the second step, that is, when there must be an authentication reinforcement (enforcement). This step is intended to mitigate attacks such as spoofing (impersonating another subject, software component, or device, mimicking their physical [71,72] or network characteristics [73,74]) and sybil (an attack where a node pretends to be several nodes [75]. This is particularly problematic in [76]) sensor networks, among others. Since the query in the second step will be redone, it is practically impossible for a malicious (real or virtual) device to “guess” the answer (CR factor) and remain authenticated to participate in the system.

The enforcement process starts when the data service (dataServ) receives data from the authenticated device. dataServ queries authServ to see if there is a need for enforcement. If so, a new query is sent as a ‘piggyback’ when returning data publication. If there is no need for enforcement, the query field will return empty. Once the query is received, $dev_{\mathrm{reg}}$ must answer it correctly or it will have its authentication expired by the authInspector and be dropped to the compatibility level.

Additionally, it is important to mention some aspects of the enforcement algorithm (Algorithm 6). This is a server-side algorithm whose function is to revalidate the access of a device that potentially committed faults, for example. Note that this algorithm starts when the reputation score is below the LOWER_THRESHOLD (Equation (Equation 4)), line 3. Similar to the authentication process, a query is generated and sent to the device (lines 4 to 8). When this is received back by the server (line 11), there is again a validation of its response, which, if positive, reinserts the device as authenticated in two factors and with an initial reputation (lines 14 to 16). However, if the query answer is not correct, authInspector will remove the device’s access to the system (line 19).
**Algorithm 6:** Enforcement Process—server-side
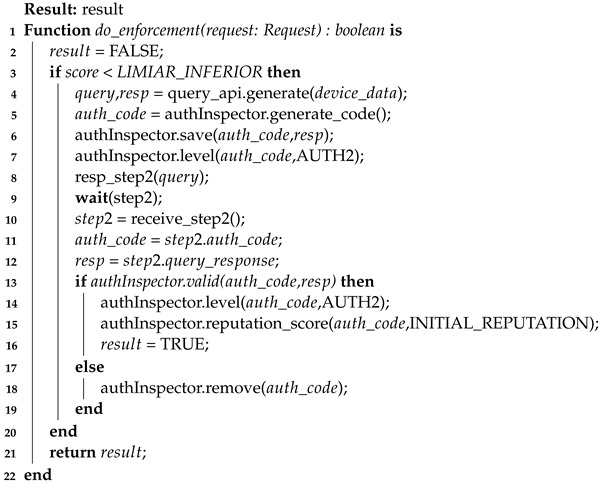


#### 2.3.4. Data Publication Process

During data publishing, $dev_{\mathrm{reg}}$ sends data packets (Equation (Equation 5)) to dataServ. Upon receiving the data, dataServ evaluates the auth_code together with the authInspector and checks if enforcement is necessary. If there is no enforcement, then four paths are possible: (i) when the AUTH3 authentication level is activated and the value is within the standard deviation; (ii) when the authentication level is AUTH3 and the value is considered invalid; (iii) when the authentication level is AUTH2 it is within the standard deviation; and (iv) when the level is AUTH2 and is outside the standard deviation.
(5)data_package<data,auth_code>

In situation (i), the value is collected and becomes the average of the monitored group. The reputation value is incremented and an ‘OK’ message is sent to $dev_{\mathrm{reg}}$. In case (ii), the value is collected and used to calculate the mean and standard deviation. The reputation value is decremented and updated. If the $dev_{\mathrm{reg}}$ is downgraded to the AUTH2 level, the enforcement routine is executed for this device. Cases (iii) and (iv) have already been addressed earlier in this text.

Regarding loss of reputation, it is observed that, if an anomaly occurs in several sensors, it will not affect the reputation system. The system is based on statistical data, and, if a catastrophe occurs, all the sensors in the group will suffer variation together. The reputation engine is designed to identify sensors with anomalous behavior and send false/altered/forged data.

### 2.4. Additional Information about the MFA_R

Before detailing the components, this section discusses the theoretical complementation of some concepts necessary for a broader understanding of how the authentication mechanism works. The add-ons that require clarification are the **compatibility mode**, the **reputation concept**, **expected values**, **query API**, **authentication levels**, and **auth_code**. These concepts are not as apparent in component modeling as they are dissolved into different components.

Firstly, concerning **compatibility mode**, this authentication mechanism aims to guarantee compatibility with legacy devices. For that, it allows the registration of old devices with a flag of “device in compatibility mode”. This tag allows the logging of device data without their reputation level. As a result, if the data consumer uses it, they will know that the mechanism does not secure the data.

Second, the **concept of reputation** in this work differs from the classical concept of distributed systems. Reputation within the context of this mechanism expresses a score that grows when publications are not identified as violations, that is, within the expected standard of publications. For example, when there is an attempt to publish random or invalid data, the attempting device suffers a penalty of its reputation score (decrease); however, when it has a normal operation, its score increases, and as a result, the device gains more reliability (addition).

Third, the **default value** algorithm (Algorithm 7) can vary according to the problem domain. There are several ways to implement data validity analysis, and they vary according to the scope of the problem. For example, let us examine three different cases: temperature in a greenhouse (I), rain in a field (portion of land used for cultivation) (II), and complex data in a plant/factory (III).

In cases like temperature, for example, limits can be used if the temperature in a greenhouse (I) exceeds 1000 degrees Celsius; this is outside a normal standard.Another case involves the use of statistical analysis. It may be compromised if a rain gauge (II) indicates 100 mm of rain in a field that others indicate an average of 3 mm.Also, as in the last case (III), we can use machine learning and submit the received measurements to its sieve. This case allows us to identify which correlated variables are deviating from the pattern that was learned as normal.

Next, the **query API** serves as a challenge–response (CR) authentication factor. Therefore, the challenge is established from the sensor registration data and the data resulting from this registration process. The query API is a set of tools for slicing, rotating, flipping, or masking, causing the data not to be fully transmitted on the network during the CR factor response process.

Additionally, the query API is a piece of software shared between the sensors and the AuthInspector. Due to the fact it is present in the sensor firmware, replicating API behavior by an adversary is very difficult. Also, using data from the moment of registration, the simple copy of the firmware from stolen equipment would cause the access to the IoT system to be invalidated. As a result, the pair {log data, query API} must be present in the sensor without any access changes.

However, it is observed that the query API can be replaced according to the application’s needs. Some studies present promising substitute factors for it, such as the use of secure hardware [77,78], authentication via token [79,80,81], physically unclonable function (PUF) [82,83], and smart contracts [50,84].

Additionally, there are the **authentication levels**, which are AUTH1 (or compatibility), AUTH2, and AUTH3. As for AUTH1, only devices registered as a legacy can publish data; in AUTH2, devices log in with two steps, receive a reputation score, and can publish data; while in AUTH3, the devices are considered reliable and must periodically perform an enforcement to keep it that way.
**Algorithm 7:** Evaluation of the published value against the default value.
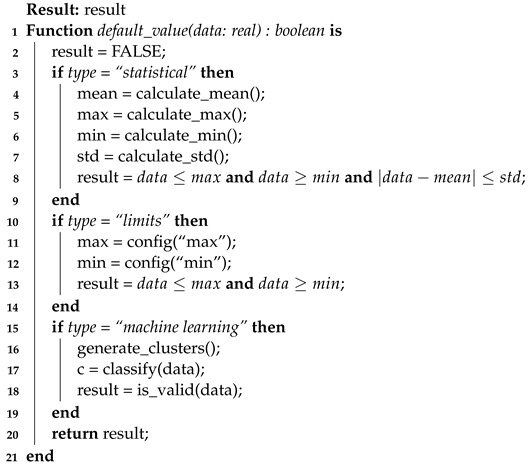


Finally, an important concept must be addressed, the **auth_code**. Such code is not an authentication token, as in token-based authentication [79] solutions. Instead, this code serves as the identifier of the authentication session established between the AuthServ and the device (components to be discussed in more detail later in this section). The auth_code is used for identifying, creating, maintaining, and expiring an authentication session for a specific device.

As can be seen, these complementary concepts enable a better understanding of how MFA_R works. Some concepts express information that does not appear in the following diagrams but is a central part of the mechanism proposed and verified in this work.

### 2.5. Authentication Mechanism Explained, Components View

For a better understanding of formal verification and how it took place in MFA_R, this section provides an overview of the components involved in its processes. Since each component has its thread and memory separate from each other, it is very important to understand how these are organized and communicated.

Although the multi-factor provides more security for authentication [85], it also involves greater complexity, and this can be seen in the various components that collaborate to enable its operation (Figure 3) in our platform. More specifically, the restricted device firmware communicates with three components on the server side: the registry service (**RegServ**), the authentication service (**AuthServ**), and the data publishing (**DataServ**). Server-side components collaborate with the authentication inspector (**AuthInspector**) and keep the data in the device data log artifact (**device_data**).

**RegServ** is responsible for registering devices in the authentication mechanism. Registration needs to be performed only once, preferably before the equipment goes into the field, using a robust network (i.e., Wireless Fidelity—WiFi) and through a secure interface (such as HTTPS). This procedure reduces the number of data transferred by the link LPWAN (it is important to note that, if the device is configured in compatibility mode, its login and password are not generated but informed at the time of registration. The registration of legacy devices occurs through a different interface from the others), which is less secure and has a lower transmission speed.

Thus, when registering, a device sends its data to the gateway, which registers it in the device’s data artifact (**device_data**). After registration, the login and password for the device are generated by RegServ and sent to the device in the registration confirmation reply message. Also, the confirmation response sends the timestamp and the registration data. These will be used as data sources by the query API in device authentication later on.

**AuthServ** is the component responsible for device authentication, queries, and enforcement. Initially, this component is responsible for authenticating the first factor and identifying the device type—in other words, if the device is in compatibility mode (a legacy device still in use) or if the device implements authentication with MFA_R. Thereafter, one of two paths can be taken: compatibility mode or normal authentication flow.

If it is in **compatibility mode**, the data published by the device will be available for consumption. All devices in compatibility mode have their published data with a reputation of zero. This behavior is due to the mechanism identifying them with great potential for data falsification. However, the consumption of this data can occur if a system does not consider this fact as problematic, or even at a time of transition in the incorporation of this technology in new sensors.

If the device is not in compatibility mode (**normal stream mode**), *AuthServ* will generate a query using the query API and send it as a second factor. Thus, the requesting device, which has the mechanism’s query API implemented, will resolve the sent query. When this happens, the device will authenticate itself with the second factor, going to the AUTH2 level. At this moment, **AuthServ** informs **AuthInspector** that the device is authenticated in AUTH2 and assigns it a score (reputation). The query works as a more robust challenge–response factor as it only passes a modified slice of the response. As a result, attacks are mitigated that statistically analyze the data sent over the network.

Remaining under the responsibility of **AuthServ**, there is the enforcement process (Algorithm 8). This process happens when a device has discrepancies in published data or meets established criteria (such as time or number of messages sent), which are determined through configuration. The enforcement requests that the device performs the second authentication step again and the sensor responds again to the query generated and sent by **AuthServ**. Upon completion, the device can send data again. If the device fails enforcement, it will be removed from the system, returning only after re-completing the entire authentication flow and gaining a new *auth_code*.
**Algorithm 8:** Enforcement process
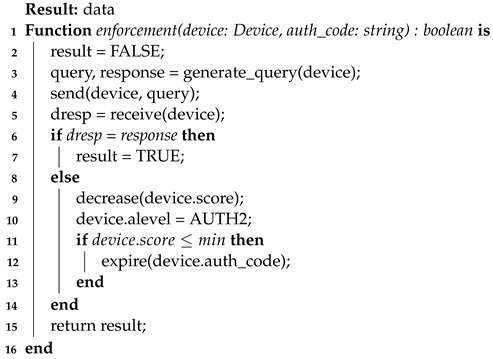


    Thirdly, **DataServ** is responsible for publishing data. The present mechanism focuses on data quality and sensor reputation, and, subsequently, authentication was implemented and proposed to guarantee security in authentication during data publication. Accordingly, to publish the data, the device must be authenticated with two factors or in compatibility mode. More specifically, when the service receives data from a device, it queries together with **AuthInspector** whether the authentication code(*auth_code*) sent is valid. Then, the **AuthInspector** can respond with three distinct values: *ok* (data can be received), *enforcement* (access needs to be renewed), and *auth_code* invalid. In the latter case, an audit log will be generated, recording the attempt to publish data improperly.

Continuing with the **data service**, it is also responsible for evaluating (Algorithm 9) whether the published data are within valid values. Once a random or invalid pattern is identified in the data reception, the device is penalized with a reputation decreasing for the device that sent the data. If the device reaches a minimum reputation level, the data service requests an enforcement to **AuthServ**. Similarly, the more values there are within the expected pattern of a device sending data, the more its reputation in the system will increase.

More specifically, **DataServ** will evaluate the data pattern and increment the device reputation up to the configured upper limit. Upon passing it, the device will ascend to AUTH3; it will be authenticated with the third factor. If the device in AUTH3 behaves inappropriately, it will suffer reputation penalties and eventually drop to the AUTH2 level. Ultimately, if it does not successfully pass enforcement, the device’s access will be removed (auth_code expires).

To access the AUTH3 level, a new type of factor is used. We call this factor **PART-WHOLE** because it is not based on something exchanged with the device but on its functioning as part of the whole IoT system—as part of this mechanism. This innovative authentication factor was proposed to identify a real part of the whole, a real part of the system. The factors in this proposal identify the knowledge of a secret, the possession of an artifact (software—API artifact), and the appropriate behavior for a real part of the array where it is inserted.
**Algorithm 9:** Process of analyzing data sent to the data publishing service
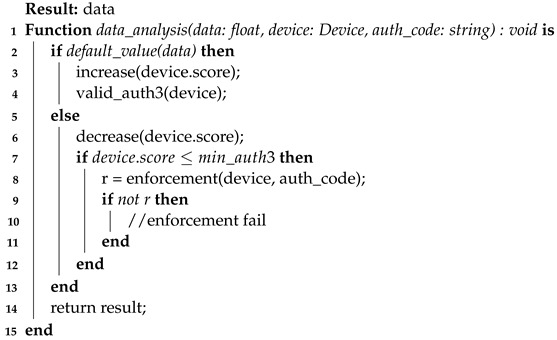


As the last part of the gateway components, there is the **AuthInspector**. Even though it has already been addressed earlier in this document, it is convenient to list its responsibilities: (i) keep a record of all authenticated devices and their authentication levels; (ii) keep a record of the sensors’ reputation; (iii) generate logs for auditing attempts at unauthorized access; and (iv) execute enforcements. This artifact is a component accessed by other components on the server-side; it does not have an external interface for access or query, and it was designed without coupling so that it does not depend on the other components. Consequently, the dependency is only in the sense of services for the **AuthInspector**, which allows for the implementation or extension of services in a simple and impactless way.

In general, all the components presented cooperate for the functioning of the authentication mechanism in their different responsibilities. Since some components will be accessed only once during the device’s life cycle, as in the case of **RegServ**, others will have recurring access, and with each publication of data, they will interact with each other. The proper functioning of the authentication mechanism lies in the responsibilities divided between the three services listed above and the **AuthInspector**. However, it is observed that some process descriptions made in this section are complemented by the state machines presented later in the next section.

### 2.6. MFA_R State Machines View

In this section, state diagrams for the MFA_R processes are presented. The processes discussed here are fundamental for the description of the behavior of the mechanism in question and serve to better understand the way in which the states evolve separately between the different components of the system.

Such state modeling is fundamental since state machines express the dynamics of the internal states of software [86] processes and components. Thus, this modeling allowed us to trace the evolution of processes and components during authentication in its different factors. Furthermore, it allowed the expression of restrictions (guards) for the evolution of states within the authentication process, which would otherwise be very complex to explain.

Moreover, this section presents the behavior of software components participating in the mechanism. Therefore, four STMs are presented: **(i)** the registration process; **(ii)** the three-factor authentication and reputation system; **(iii)** the sensor life-cycle; and **(iv)** the evolution of authentication levels. By following the evolution of these STMs, it was possible to visualize the evolution of each process and of the software component within the mechanism. Consequently, this fact allowed us to refine state modeling and MFA_R.

It begins with the **(i)** registration process (Figure 4), where the system represents the device through three states: not registered (*n*), in registration (r0), and registered (r1). The evolution between the states occurs through the registration request sent by the device (reg), which can be accepted (reg_ok) or not (reg_fail). If accepted, the device becomes registered and leaves this state only when the unregister process (unreg) is requested. Consequently, once registered in the mechanism, the device can perform the authentication process and the subsequent publication of the data described later in this document.

Presented in **diagram (ii)** (Figure 5) is a unified overview of all processes involved in authentication. It describes the processes of one-factor authentication, two-factor authentication, data publication, three-factor authentication, reputation system, and possible cases of failure in the process, which are further detailed below.

**First factor** authentication begins with a message sent to **AuthServ** containing the device credentials (step1). Then, validation is performed by **AuthServ**, generating the event of a valid password (step1_ok) in the case of success. If there is an authentication failure, the mechanism will evolve the device representation to a critical failure state (q8). If the first-factor authentication is successful, the device can evolve to an authenticated state in compatibility mode (q4), or the mechanism can generate the query, sending it to the device (q2↦q9↦q2). As for the next steps, once the query is sent along with the *auth_code*, the authentication mechanism will wait to receive the response from query to continue evolving.

As for **factor two**, after receiving the response from query (step2), the mechanism evolves to its validation state of query (q3). If the query answer is wrong, the STM goes into a critical failure state (q8). If it is correct, the mechanism can wait for the publication of data (q5), which must occur through **DataServ** using the *auth_code* obtained by the device.

Once in the data waiting state (q5), already at the AUTH2 authentication level, the reputation system starts to be used. First, the device receives an initial reputation value (score), whereby the data publication evaluation process will be maintained. Thus, after the correct data publications (q11), the score is incremented (increase_reputation). Similarly, the reputation score will be decremented (decrease_reputation) if data publications are outside the configured standard (q7). As a result, following successive valid publications, the device reputation will pass the configured upper limit (score>max), and the device will be eligible (raise_authlevel) to level up (q12) via the eventstep3_ok.

Additionally, as fault flows one can list (a) the reception of an invalid *auth_code* and (b) the reception of invalid data. As such, when receiving an invalid *auth_code* (a), the mechanism invalidates the access (authcode_fail) and evolves to the critical failure state (q8). On the other hand, when invalid data are received (b), the device’s score is decremented. After successive publications of invalid data, the score will reach the lower reputation threshold, resulting in a request for enforcement. Also, if the enforcement is successful, the device will send data back to the system but with the authentication level AUTH2. However, if it is not in AUTH3 but rather AUTH2, the device will have its access invalidated (invalidate_access), reverting to the critical failure state (q8). Consequently, if enforcement fails, its state will devolve to critical failure (q8), and the device will exit the system.

The next diagram is the **Sensor STM** (iii), which demonstrates the device lifecycle (Figure 6) and reflects the influence of the evolution of the authentication mechanism on the sensor. Even though some aspects of the sensor have already been covered in other diagrams, it is important to point out some facts discussed below. The sensor starts its life cycle without being registered (q0) and finalizes it in two ways: through critical failure (q2) or after its data-sending cycle (q7). It is worth noting that the device will only send data after successful second-factor authentication (step2_ok).

It is also noteworthy that compatibility mode operation does not have a representation of internal states. Even without a representation of internal states, the mechanism needs to know all devices that are working in compatibility mode (q8).

Lastly, there is the **authentication levels** STM (iv), which represents the evolution between the authentication levels (Figure 7). During the operation of MFA_R, its main objective is to more clearly show the essential steps for evolution in authentication levels and their order of occurrence. Also, it transparently represents the main failure flows that lead to a critical failure (q4). Subsequently, we can cite: failure in the first factor (step1_fail), failure in the second factor (step2_fail), failure in *auth_code* (authcode_fail), and access invalidation (invalidate_access).

Although a minor detail, it is important to note that there is no transition from failure to the third factor (reputation failure). A failure of this type can lead to enforcement, and it can result in a failure of the second factor during the enforcement process. As such, when authenticated at the AUTH3 level, a device eventually is removed from the system if it does not perform an enforcement correctly. This action prevents temporary read errors from taking the device out of the system.

With the use of STMs, it was possible to detail the dynamic aspect of some components of the mechanism. Such STMs provided the necessary level of detail for the construction of automata, which support the formal validation of some properties of the mechanism.

### 2.7. MFA_R Timed Automata Modeling

This section presents the formal modeling of MFA_R using timed automata. The automata provides an option for a formal approach that can be used to model dynamic aspects and their variability, going beyond the expression potential of the UML [87]. As a result, this is a formal approach that guarantees formal verification by confirming the (safety) properties of the system.

To ensure such properties, the tool used in the modeling was Uppaal. This tool is the de facto standard for modeling timed automata [88], through which it was possible to express the temporal evolution and synchronize events through channels, use variables to control the flow, and perform the simulation of the designed models. In addition to the functionalities mentioned earlier, the programming of functions is a possibility that the tool brings and that allows us to extend its potential.

Additionally, with Uppaal, it was possible to model several timed automata to evolve in parallel [89]. This design was possible because a project in this tool can describe different templates, and each template represents the modeling of an automaton. Consequently, in order to be simulated, a template must be instantiated in the system declaration section of the project, and every template can be instantiated as many times as necessary. For example, to model the behavior of five identical components, create a template and instantiate it five times.

It is still possible to communicate the templates via channels (channels-chan) and chain events between the templates. However, as mentioned earlier, the tool also provides an interface for queries using temporal logic with reduced TCTL. As a result, it was noted that Uppal was the most suitable choice for developing automata models.

Thus, in the present study, four models (templates) were created to represent the components and processes of the mechanism. Such a creation was based on the previously presented state machine diagrams. Comparatively, due to the greater representativeness of the ATs, it was possible to express in the latter the dependence between the events that happen in different components (synchronization), unlike the pure STMs. Consequently, this dependency allowed us to chain events and to model and verify the system’s evolution and componentsall concurrently and in sync.

The first model is the **sensor registration** process (Figure 8), which contains three states and four transitions. The states represent the unregistered device (*n*), the device awaiting registration (r0), and the registered device (r1). It is concluded that the diagram is very similar to the STM that gave rise to it (Figure 4), asserting the transparent diagram conversion process and lifecycle monitoring through TAs as well. It is important to remember that this phase of the authentication process takes place over a network with higher data throughput than the LPWAN and a secure channel (i.e., HTTPS). Although this phase is safe from the network’s point of view, its modeling is necessary for the feasibility of other properties.

The second diagram (Figure 9) models the sensor’s **authentication process**. Namely, it models the behavior of the authentication mechanism as a whole. In general, this model is the core of the proposed authentication mechanism. More specifically, it expresses the three authentication factors, the reputation system, and access invalidation. This model is responsible for triggering most of the transitions that will allow the other templates (models) to evolve. Its design phase was the most difficult phase of this work and took several cycles, which resulted in the refactoring of the related STMs.

In modeling the **sensor** (Figure 10), it was decided to establish the beginning of the model from the device already registered (q3). This decision was made to allow the isolation of only the part of the mechanism that performs the authentication process during the simulation. Only the LPWAN part of the system in question was simulated. In turn, this model is responsible for triggering important events such as starting authentication, sending data, and resetting. This last event is linked to the device’s hard reset button and causes a failed sensor (removed from the system) to restart the authentication process.

As a last model, there is an overview of the **authentication levels** (Figure 11). In general, through this model, it is possible to follow the evolution of the authentication mechanism in its authentication levels and visualize its main flaws. Also, it should be noted that this model does not trigger events but serves as an overview of the mechanism. This overview makes it possible to propose a clearer visualization of the sensor states. In general, it would be possible to implement a human–compter interface (HCI) panel for debugging, for instance.

As a result, this set of models allowed us to build a basis for formal verification as the use of TAs enabled successful verification of properties. Additionally, it promoted a better understanding of modeling details that evolved during the present study. Correspondingly, it was possible to simulate the set of models built through this evolutionary process and later query the properties of the—model, which are better explained in the next section.

## 3. Results—Formal Verification of Security Properties Evaluation

This section presents the simulation (Figure 12) of the MFA_R mechanism and its formal verification through the TCTL queries performed in the selected tool. Such tasks were conducted through Uppaal’s simulator tab, a built-in tool that allows for the visualization of available transactions, lists the steps taken during execution, and enumerates the variables list. Furthermore, through these tools, it is possible to visualize the instantiated templates and a diagram showing the exchange of messages during the simulation (this can happen step-by-step or automatically since, in the automated case, the presentation speed of the transitions can be configured). Additionally, evaluation through a trace log (simulation trace) of the paths that lead to unwanted states (i.e., deadlocks) is permitted. This feature allows us to correct them at design time, preventing future crashes. This simulation was developed using an Intel Core i7-4510U processor and 16 GB of RAM with the system Ubuntu 18.04.03 LTS operating.

As for the verification of properties, the security aspects (Table 6) were converted to the TCTL query language, according to Table 8. Thus, the conversion did not take place directly (one-to-one). Some aspects were converted into more than one query to provide a complete verification of the aspects.

First, it was analyzed if the model is free of deadlocks (Table 8—**c00**), and for that, a query was executed that evaluates each possible execution path (A[]). The result was that there are no deadlocks in the project, and this result was made possible through some refinements performed during the development of the study. One of the consequences of this verification proposed here was the correction of the initially proposed behavioral model. As a result, there were modifications to the STMs and ATs, which made the MFA_R deadlock-free.

As for the **a01** aspect, two queries were created: **c01** and **c02**. The first one (**c01**) checks if there is a possible way to reach the data reception state (authentication.q5) without going through the positive result of the second-factor authentication (authentication.q3). The second query (**c02**) can be translated as evaluating whether there is a path that allows the sensor to send data (sensor.q7) without the authentication mechanism being ready to receive (authentication.q5). The first validation was presented as TRUE, and the second was FALSE. Consequently, the combination of the two leads to the conclusion that sending data without going through the second authentication factor is impossible.

Following the **a02** aspect, it was verified whether the discovery (spoofing) of *auth_code* leads to removal from the system. This aspect was translated into the query (**c03**), which checks if an *auth_code* error is identified at any time; the mechanism will eventually end up with a critical failure (authentication.q8). This query is TRUE and used the error code generated by the authentication template. As a result, it was impossible to spoof/discover auth_code without being discovered and removed from the system.

Also, the verification of the **a03** aspect analyzes credential theft and has a direct correlation with **c04**, its verification-returned TRUE. Therefore, the TCTL query (**c04**) can be translated as follows: when presenting an error in enforcement, the mechanism generates a step2_fail and marks the device with an error code STEP2FAIL. Its mitigation occurs because even if an attacker steals valid credentials, he will be discovered at the time of access renewal when executing an enforcement. Consequently, when failing enforcement (ste2_fail), the device will have its access invalidated.

Finally, there is the aspect that addresses the publication of random data (**a04**). Once the receipt of invalid data (data_fail) has been identified, two possible endings will occur: DATAFAIL or STEP2FAIL. If the device does not have enough score and its AUTH2 level, it fails with DATAFAIL as an error code. However, if you have enough score and it is level, an enforcement may be requested, and in case of violation it will fail (step2_fail). In both cases, the inevitable conclusion is the evolution to the critical failure state (authentication.q8), invalidating access. The check for **c05** was already successful in the previous step, and the check for **c06** also returned true. We, therefore, conclude that publishing random/incorrect data will eventually lead to sensor removal.

All checks were successfully carried out on the model, and the following was concluded: the models evolved to become deadlock-free; it was not possible to send data without going through the second authentication factor, and all auth_code spoofing will eventually be discovered and removed from the system, similarly stealing credentials and publishing random data. Thus, through formal verification using this tool, it was possible to assess that the TA modeling covered all security aspects proposed by the MFA_R.

## 4. Discussion

This section presents some considerations, the secondary results, the results of each workflow step, and the simulation results. The secondary results were unexpected developments in this work, which are important for the conclusion and result. The results of each stage allowed us to follow the evolution of this work and verify its correctness throughout its development. Finally, the simulation results are the final results of the entire workflow, the result of evaluating the listed properties.

Regarding the **considerations**, although there are different approaches to modeling and converting UML diagrams to [90,91,92] automata, we opted for manual conversion in this work. This conversion was carried out based on the life cycle projected by the states of the UML and expressed through the evolution of states in the AT. Some works prove the correctness of this method [93,94], for example, through bisimulation between the converted models. This choice was based on the team experience capacity for choices during the desired conversion.

Regarding the adoption of timed automata, this approach was necessary for quantifying some attempted actions during authentication (i.e., publications with invalid data). This type of quantification would bring additional complexity to other approaches, mainly in validating the desired properties in the model. Additionally, the correlated works reinforce and validate the importance of using timed automata for this situation.

Although there are other highly relevant formalisms, such as Petri nets and colored Petri nets, the experience and set of previous works developed with TAs led us to adopt this approach in this study. Furthermore, ATs is a recursively enumerable language belonging to the same Chomsky classification for colored Petri nets [95], with the same expressiveness. Therefore, the choice of approach becomes a design decision only, and in this project, we decided to follow our expertise with TAs.

Some **secondary results** are refinements of the state diagrams caused by the challenges encountered during the simulation. For example, in the simulation process, it was observed that the automaton went into dead-lock when the device was unauthenticated; yet, in some situations there was an interlock between sending data to the sensor and the mechanism. These challenges caused modifications to the STMs, leading to a modification to the ATs. As a result, the mechanism became deadlock-free. Another important secondary result was the verification and evaluation of the distributed nature of the mechanism’s components. Through the simulation, it is possible to visualize the signaling between the various processes involved and prove the need for a formal verification that addresses temporal properties, such as the non-interlocking of the processes.

As per **results of the workflow steps**, first, there is the section of **related works** where 20 different works were considered that used four different approaches of temporal logic for their accomplishment. Since most of the works were published in the last two years, it is possible to conclude that formal verification is a current and adequate tool for evaluating security solutions. This fact happens in a way that complements the security verification of protocols (i.e., AVISPA), going beyond the verification of information aspects and the availability (non-locking) of the proposed solutions.

Analyzing the data from the related works, Figure 13a,b, it is possible to access some important data. In regards to the annual distribution, Figure 13a, the works are concentrated in the last two years (2020 and 2021), around 75% of all correlates. Still, on the annual distribution, the oldest work is from 2004 and uses LTL as the chosen approach. As for the approach used, Figure 13b, TCTL has the largest number of correlated works. This data confirms this approach’s importance for the formal verification of works related to the security area.

Regarding aspects of advantages, disadvantages, and limitations, the works presented here exhibit a relevant view of the area of formal verification and security, in particular those focused on the area of authentication and IoT. However, even though they put forward a great contribution, none presents an intelligible and easily replicated view of the construction process of the artifacts involved in the modeling necessary for the formal verification of the security properties of the proposed solutions. Our work also contributes through this replicable process, which ranges from the selection of security aspects through threat modeling to their conversion and verification through temporal logic using timed automata.

As for the **TM** presented, it contributed to identifying threats that should be evaluated. A more detailed view can be obtained by analyzing the artifacts published on our project website (https://github.com/wesleybez/mfar_formalverification, accessed on 1 July 2023) and in another publication by the authors that describes the threat-modeling process in greater detail. Furthermore, its process took place in three stages: the definition of the attack surface; the identification, categorization, prioritization, and risk assessment of threats; and the description of countermeasures through the security requirements shown in Table 6. As a result, threat modeling proved to be a thorough assessment that produced important artifacts for formal verification.

Quantitatively, the data on threat modeling can be seen in Figure 14a–c. Figure 14a shows the number of threats by STRIDE category, highlighting the tampering (T) category that had the highest number of threats and the repudiation (R) category with zero threats. In Figure 14b, which shows threats by attack surface, the negative highlight goes to s05 (LPWAN), which is the AT most susceptible to causing more damage, according to DREAD. Finally, the DREAD methodology presents v7 (tampering data) and v11 (unauthorized access) as the threats that create the greatest risk to the mechanism.

The **components** came with an overview of the parts of the authentication mechanism and how they communicate (with each other and with the devices). Additionally, they explained how MFA_R works through descriptions and algorithms, explaining its operation and clarifying some of its details.

As for the **STMs**, it can be noted that the diagram was generated from the authentication STM (Figure 5) but has significant changes, such as the removal of compatibility support, the possibility of sensor reset, and the abstraction of query generation. However, it is important to point out that the modeling did not happen linearly. After verifying the needs of new states in the TAs, it was necessary to go back to the STMs and correct them. Accordingly, the proposed process helped refine the MFA_R modeling, allowing one to correct problems at design time.

As for **automata modeling**, it is important to highlight three points: (i) **compatibility mode**, (ii) **sensor reset**, and (iii) **query abstraction**. Therefore, the diagram does not cover support for **compatibility mode** (i) as it is understood that the decision to operate a device in compatibility mode is prior to authentication itself. By incorporating the compatibility state into the model, the automaton would be in a livelock, which can denote starvation or infinite execution of a state cycle [96]—it never reaches a final state, an infinite movement between states [97], or deadlock, which are associated with threads that are waiting for events to continue executing [96], usually associated with the use of shared resources; this would produce problems for validating other properties—if simulated and evaluated in the project.

The option of **reset on the sensor** (ii) allowed us to remove the deadlock in case of invalid access. The model can generate a reset event from the sensor and cause the other model instances to restart their functioning. This event can only happen after invalidating the access by the authentication mechanism (q8). Thus, it is important to point out that, after invalidating the access, the sensor’s reset is the only event the model can execute, allowing it to interact with the authentication mechanism and with the **DataServ again**.

The **abstraction of the query** (iii) did not lead to any problems during modeling. This is due to the fact that, after generation, the mechanism (Figure 9) expects to receive the second authentication factor with the response of query (step2). Therefore, due to the cause and effect relationship between the generation and the response of the query, it is considered that the events are already implicitly attended by the model.

By quantifying the effort made in the modeling, Figure 15, it can be seen that authentication was the one that demanded the most effort in this work. This representation of the authentication process takes place internally in the gateway. It is almost twice as complex as its counterpart in the sensor in both the STM and the TA. In turn, registration is the simplest of all four processes; however, it is of great importance as it is through this process that sensitive data are transmitted and the credential (login/password) is created.

No less important was the modeling of queries in TCTL. This step allowed us to simulate a complex system with distributable components. Thus, it allowed one to verifying ownership at design time to guarantee the security of the authentication mechanism using modal logic.

As a result, the work presented a formal verification that ensured the properties that describe the security requirements arising from the TM performed. However, it was not limited to this, also presenting a modeling of components, their dynamic aspects through the STMs, and the incorporation of temporal characteristics to these with the TAs all this in a well-documented and reproducible way in the work of future researchers.

## 5. Conclusions and Future Works

The work successfully and formally verified, through ATs and TCTL, the correctness of the proposed mechanism. It was verified that the mechanism is free of dead-locks (c00), as well as the condition of being authenticated to send data (c01 and c02), and, finally, that failures in authentication, sending data, and impersonation attempts lead to a final state of failure (c03–c06). Such correctness translates into the successful verification of properties created from security requirements. With a workflow that covered everything from components to modeling in timed automata, it was possible to create a framework of artifacts that supported it. Thus, through formal methods, the properties of models were successfully demonstrated, which expressed the safety requirements.

In addition, a set of reusable resources was created for the extension or creation of a mechanism similar to the one proposed (project site (https://github.com/wesleybez/mfar_formalverification, accessed on 1 July 2023)). Accordingly, a survey of the state-of-the-art and related works was carried out; threat modeling was also designed that supported the selection of security requirements (aspects) that would be transformed into TCTL properties, following the modeling of components and a short description of its operation; and the modeling of state machines, and their conversion to timed automata was carried out, ending with the creation of properties in TCTL and their verification. The entire process was self-explanatory so that it could be easily verified and reused in future works. As such, any work of similar scope will be able to follow the workflow proposed here and one can use the artifacts published on our GitHub as a basis for producing one’s own.

Overall, formal verification was used to demonstrate the properties of our reputational multi-factor authentication mechanism for restricted devices and LPWAN. Thus, it was possible to express the evolution of the component states during the mechanism operation and model them so that there were no interlocks between the parts. The security aspects that the mechanism meets were also successfully verified.

In future work, some of the other security aspects to be complemented in the model will be analyzed. Verification will also be performed using tools that use the Dolev–Yao [98] model of an adversary to validate security protocols (i.e., ProVerify and Tamarin Prover). This additional validation aims to better understand the aspects involved in the communication and evaluate the communication protocol used in the data exchange between the devices. Since timed automata are not capable of fully analyzing some aspects of the cryptographic protocols involved in LPWAN communication, the use of such verifiers becomes very important.

Additionally, reinforcement in the present study will be carried out through a more in-depth analysis of the threat model applicable to the proposed mechanism and the mitigation of threats found. This becomes essential when the network technology to be used at the time of analysis is brought to light. LPWAN protocols, to a large extent, add a layer of security to the system as it already has some security features implemented in its protocols.

Finally, a more practical study of the MFA_R application is also necessary through scalability issues, deployment options, and simulation using real sensors—going beyond the computational simulation carried out in the previously mentioned articles [14,28,29]. 

## Figures and Tables

**Figure 1 sensors-23-06933-f001:**
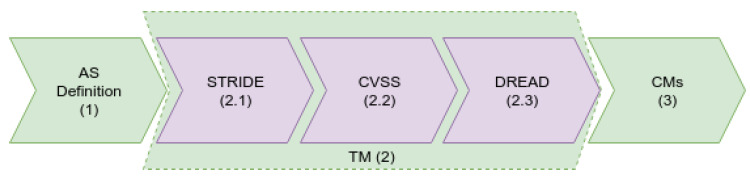
MFA_R Threat-modeling steps—in green, the macro steps are detailed; in purple, the three threat analysis (TM) steps are detailed. The workflow was divided into three macro-steps, with the second step being detailed in three other sub-steps.

**Figure 2 sensors-23-06933-f002:**
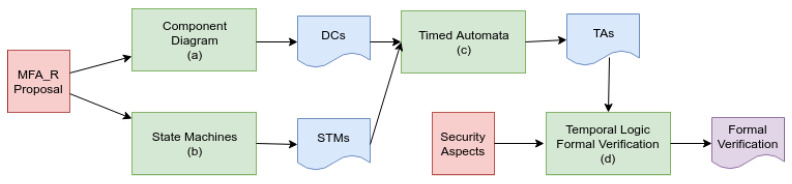
Steps to develop the study for formal verification of the proposal—existing artifacts are shown in red; steps developed and reported in this study are shown in green; artifacts produced as a result of these steps are shown in blue; and the final result of the work is shown in purple.

**Figure 3 sensors-23-06933-f003:**
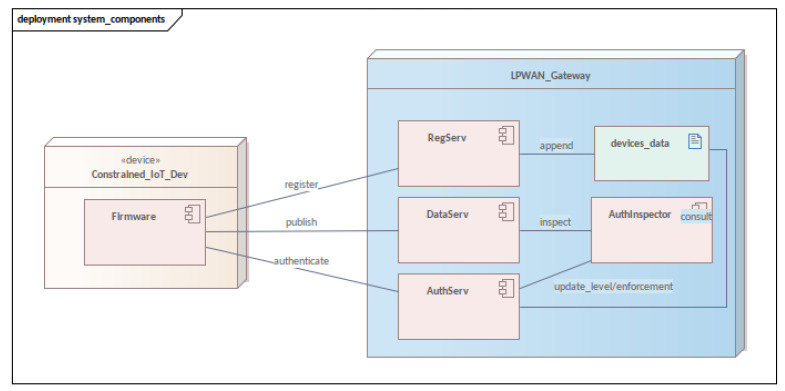
Component diagram with main components of the multi-factor authentication mechanism with reputation—on the left is the node representing the IoT restricted device, and on the right is the LPWAN gateway with the authentication mechanism components.

**Figure 4 sensors-23-06933-f004:**
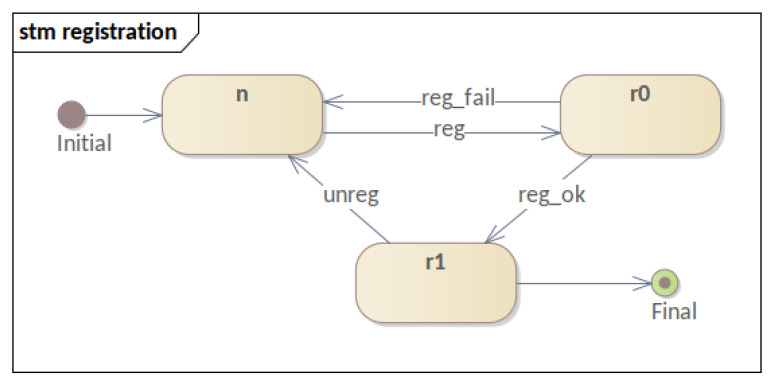
Process of registering a device in the proposed mechanism. The diagram is composed of three states and four transitions, and its function is to model the evolution of device states against RegServ.

**Figure 5 sensors-23-06933-f005:**
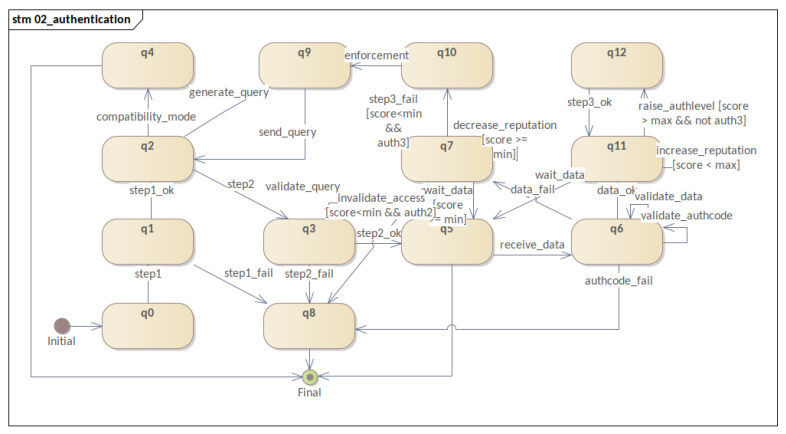
State machine for three-factor authentication, enforcement, and data publishing. The diagram is made up of 13 states and more than 25 transitions. It is the most complex state diagram of this work and the heart of the MFA_R operation. It represents the main processes and points for exploiting the listed vulnerabilities.

**Figure 6 sensors-23-06933-f006:**
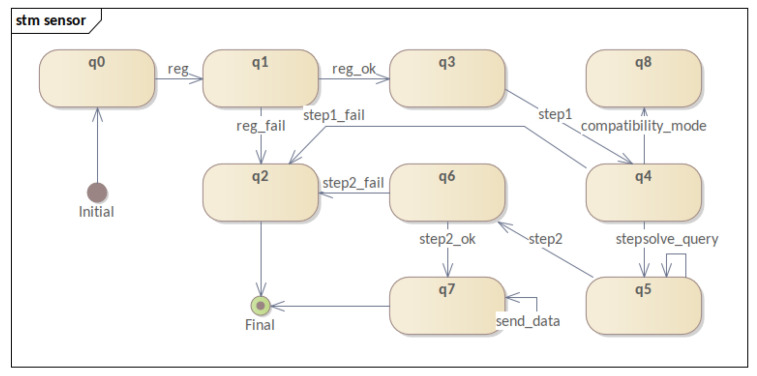
Sensor lifecycle state machine. It consists of eight states and 12 transitions. The diagram represents the sensor’s view of its internal state concerning the system and authentication mechanism.

**Figure 7 sensors-23-06933-f007:**
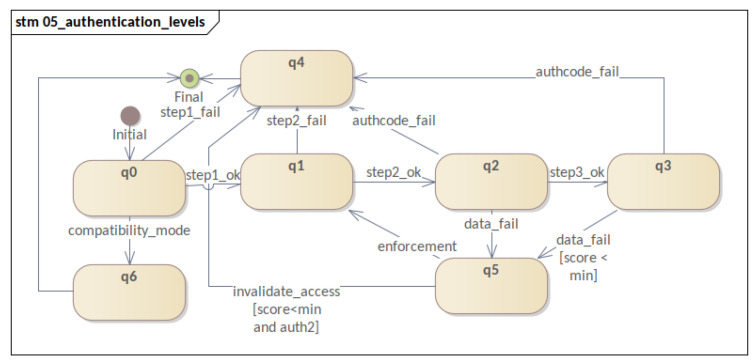
The evolution of the authentication levels’ state machine diagram for the proposed mechanism. It features seven states and 12 transitions. It is responsible for modeling the behavior, evolution, and constraints of MFA_R authentication levels.

**Figure 8 sensors-23-06933-f008:**
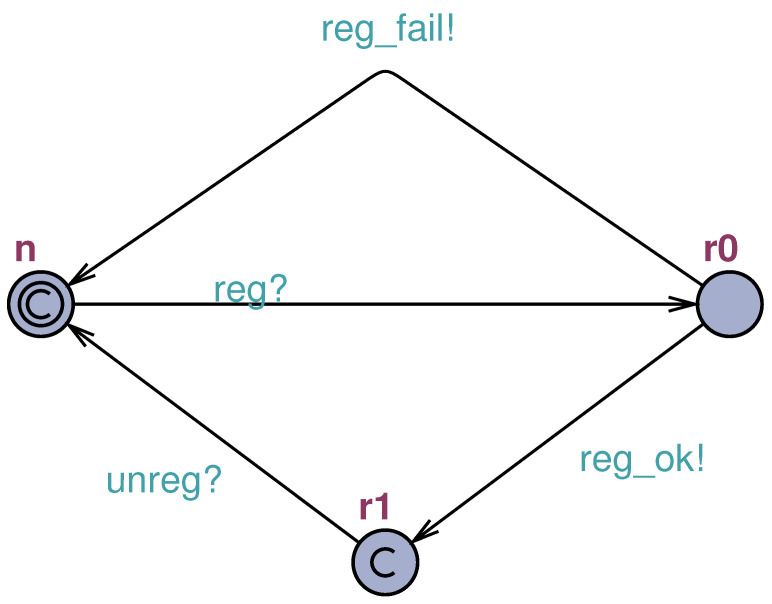
TA of the sensor registration process. Composed of three states and four transitions, it has two committed states, two transitions that generate events, and two transitions that wait for events.

**Figure 9 sensors-23-06933-f009:**
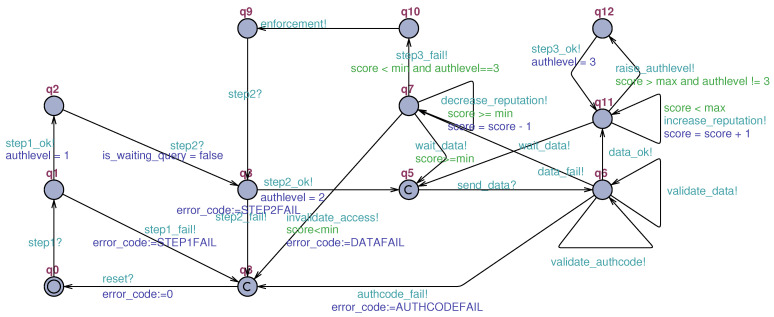
TA of the sensor authentication process. It features 13 states and 23 transitions, including two final states, 15 event generations, and eight event queries. It controls and communicates with the other models (templates) to coordinate the operation of MFA_R.

**Figure 10 sensors-23-06933-f010:**
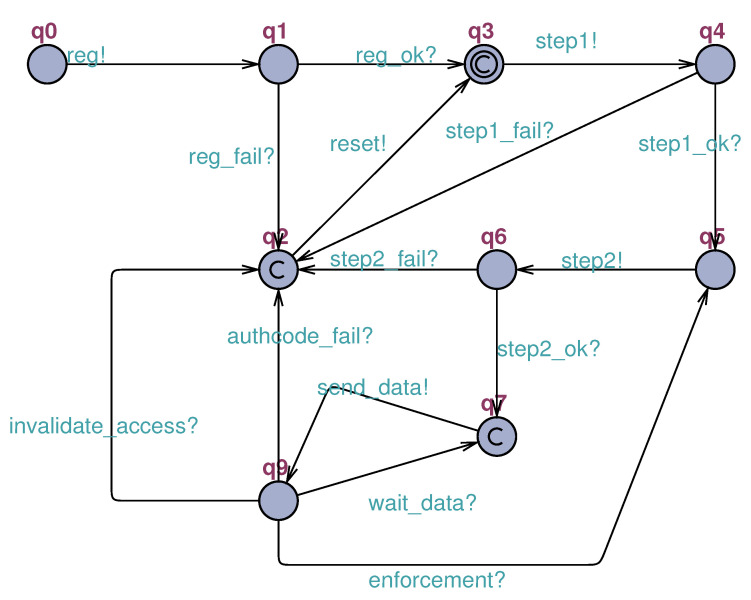
Sensor lifecycle TA. It has eight states and 14 transitions, three final states, four event generations, and ten event queries. As a need for simulation, this diagram incorporates aspects of registration, authentication, and publication of—data, always from the sensor’s point of view.

**Figure 11 sensors-23-06933-f011:**
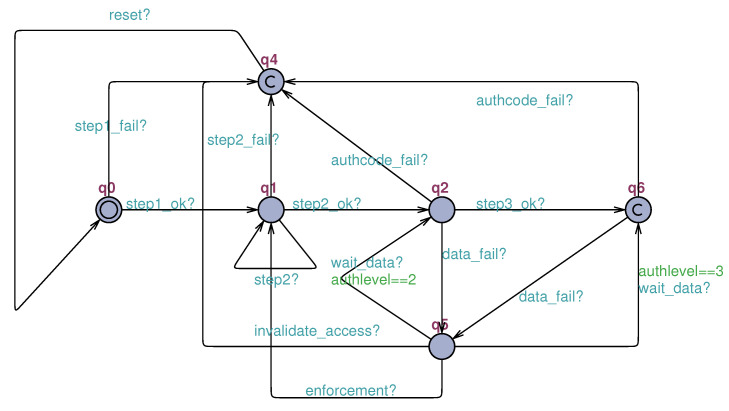
TA of the evolution of authentication levels within the proposed mechanism. It contains seven states and 13 transitions, two of which are final states and all-event query transitions. It serves as a panel for better understanding and priority queries of the other templates.

**Figure 12 sensors-23-06933-f012:**
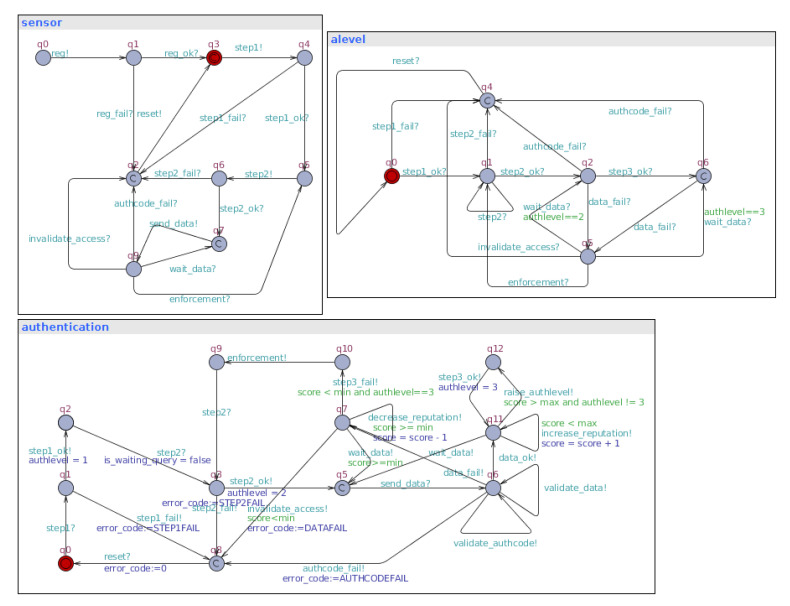
Simulation of models through instances within Uppaal—in this simulation, we can see an instance of a sensor (sensor), one of the authentication mechanism processes (authentication), and one of the authentication levels (alevel).

**Figure 13 sensors-23-06933-f013:**
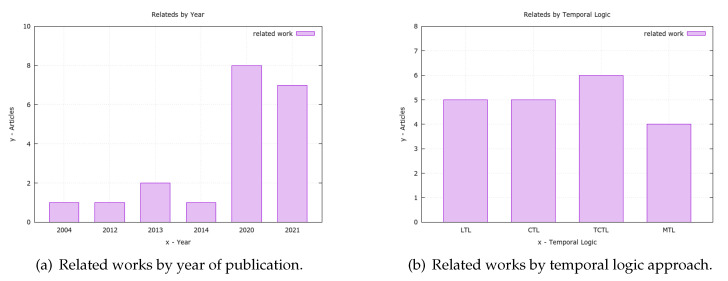
Related works—quantitative data.

**Figure 14 sensors-23-06933-f014:**
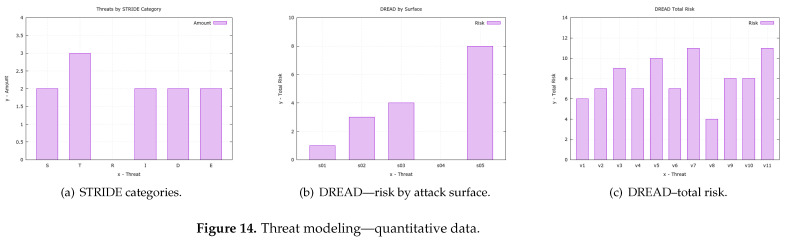
Threat modeling—quantitative data.

**Figure 15 sensors-23-06933-f015:**
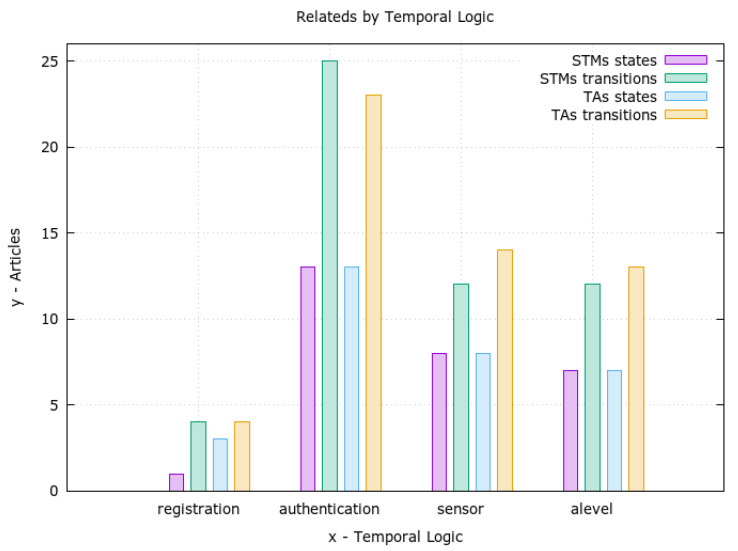
Quantitative overview of state diagrams and modeling in timed automata.

**Table 1 sensors-23-06933-t001:** Works that correlate security and temporal logic—the works are organized and presented by a temporal logic approach. The works are sorted in descending order according to the year of publication.

#	Articles	Application	Year
LTL			
p01	Kuze et al. [36]	UAV	2020
p02	Salva e Blot [33]	IoT/11 measurements	2020
p03	Tun et al [34]	Weakening user obligation	2020
p04	Ouchani e Debbabi [35]	UML e SysML	2020
p05	Grosu et al [32]	Edhan–Schoroeder	2004
CTL			
p06	Mbongue et al. [42]	Multi Tenant FPGA/cloud	2021
p07	Lilli et al. [43]	Z-Wave	2021
p08	Gava et al. [44]	On-the-fly checking	2014
p09	Maurya et al. [41]	Zero-knowledge Authentication/Fiat–Shamir	2012
p10	Valadares et al. [45]	Trusted architectures for IoT	2021
TCTL			
p11	Park et al. [50]	Smart contracts	2021
p12	AlQadheeb [47]	Security policies	2020
p13	Gu et al. [49]	Autonomous agents	2020
p14	Askapour et al. [51]	Robotic mission plan	2020
p15	Camilli [52]	Microservices	2020
p16	Malik et al. [48]	ICCP	2013
MTL			
p17	Ammar et al [58]	Opacity	2021
p18	Ozmen et al. [53]	Apps IoT	2021
p19	Yahyazadeh et al. [59]	Apps IoT	2021
p20	Ahmed [56]	IDS	2013
→	This work	MFA_R (TM, components, TA, STM, and TCTL)	2022

**Table 2 sensors-23-06933-t002:** Description of attack surface for MFA_R—surfaces are grouped into three categories (external actors, services, and data links). The data are organized in the surface column and the vulnerability column, with the latter being a set of all vulnerabilities listed for the related surface.

#	Group	Surface	Vulnerabilities
s01	actors	Device	impersonation, sleep privation, and jamming
s02	services	RegServ	information disclosure, leakage e DoS
s03		AuthServ	device impersonation, brute force attack
s04		DataServ	device impersonation, and tampering data
s05 s06	links	HTTPS LPWAN	information leakage, MITM message tampering, message fabrication, and unauthorized access

**Table 3 sensors-23-06933-t003:** DREAD risk evaluation—this artifact presents quantitative risk assessment through the DREAD tool. From the left is the vulnerability identification column; its brief description; the five evaluation categories of this TMM; the total sum of the evaluations; and the averages of the evaluations, that is, the risk.

#	Threat	D	R	E	A	D	Total	Risk
v01	information disclosure	3	0	0	1	2	6	1.2
v02	leakage	3	1	0	1	2	7	1.4
v03	DoS	3	2	0	2	2	9	1.8
v04	device impersonation	1	2	1	1	1	7	1.4
v05	brute force attack	1	3	3	1	2	10	2
v06	device impersonation	1	2	1	1	1	7	1.4
v07	tampering data	2	2	2	3	2	11	2.2
v08	information leakage	2	0	0	1	0	4	0.8
v09	message tampering	1	2	2	1	1	8	1.6
v10	message fabrication	1	2	2	1	1	8	1.6
v11	unauthorized access	3	2	2	2	1	11	2.2

**Table 7 sensors-23-06933-t007:** Symbol table.

Code	Description
$dev_{\mathrm{reg}}$	registered device
$dev_{\mathrm{unreg}}$	device not registered
gw	gateway, device where services are published
$db_{\mathrm{gw}}$	database located at gateway
regServ	registry service
authServ	authentication service
authInspector	authentication inspector
dataServ	data service
auth_code	authentication code
authetication_query	query used as second authentication factor

**Table 8 sensors-23-06933-t008:** Queries are built for formal assessment of security aspects against the formal model. From the left, the first column contains the query’s identification, the next shows the aspect it verifies, and the last shows the TCTL source code of the query.

#	Aspect	Query
c00	–	A[]notdeadlock
c01	a01	notauthentication.q3−−>notauthentication.q5
c02		E<>sensor.q7andnotauthentication.q5
c03	a02	A[]error_code==AUTHCODEFAILimplyauthentication.q8
c04	a03	A[]error_code==STEP2FAILimplyauthentication.q8
c05	a04	A[]error_code==DATAFAILimplyauthentication.q8
c06		A[]error_code==STEP2FAILimplyauthentication.q8

## Data Availability

Not applicable.

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
