# Peer review of "A Formal Verification of a Reputation Multi-Factor Authentication Mechanism for Constrained Devices and Low-Power Wide-Area Network Using Temporal Logic"

_sensors, 2023, doi:10.3390/s23156933_

Round 1

Reviewer 1 Report

Points to the authors

1.     
The authors are advised to define each abbreviation and notation at first use. I can see that the authors missed many of them. Please check for this

2.      The preface of this paper needs some work. I suggest the authors add some relevant information about the Internet of Things, and link it with the security.

3.      The contribution points need to be explored. Please keep in mind non-specialist reader.

4.      I suggest the authors to add this latest literature in the revised paper to increase the reader interest in this work . 1. Adil, M., Attique, M., Wang, J., Alrefaei, F., Song, H., & Farouk, A. (2022, December). IoST: Internet of Softwarized Things Networks, Security Challenges and Future Research Directions. In 2022 IEEE Globecom Workshops (GC Wkshps) (pp. 269-274). IEEE. 2. Gupta, M., Singh, V. P., Gupta, K. K., & Shukla, P. K. (2023). An efficient image encryption technique based on two-level security for internet of things. Multimedia Tools and Applications82(4), 5091-5111. 3. Kaur, B., Dadkhah, S., Shoeleh, F., Neto, E. C. P., Xiong, P., Iqbal, S., ... & Ghorbani, A. A. (2023). Internet of things (IoT) security dataset evolution: Challenges and future directions. Internet of Things, 100780. 4. https://doi.org/10.1016/j.scs.2021.103311.

5.      The visualization of all images should be improved. Presently, all of them are blurred

6.      I am also not convinced of your proposed model. Please work on this section, make it concise and understandable for the readers.

N/A

Reviewer 2 Report

While the manuscript provides a detailed analysis of a reputational multi-factor authentication mechanism for restricted devices and LPWANs, there are several areas that require improvement. 

The abstract provides a general overview of the manuscript's topic, however it lacks specific details about the proposed analysis and its significance. The authors can consider adding more specific information about the research objectives and outcomes. 

Lack of clarity in the introduction: The introduction fails to clearly outline the existing limitations and shortcomings of current authentication mechanisms for restricted devices and LPWANs. Providing a more comprehensive overview of the challenges faced in this domain would enhance the reader's understanding of the research objectives.However, by omitting in-depth descriptions of particular risks and motivations, the introduction could be made more succinct and focused.  Instead, provide a brief summary and refer readers to the related works section for further information

Insufficient analysis of related works: Although the related works section includes a wide range of references, there is a lack of in-depth analysis and comparison. The manuscript would benefit from discussing the strengths, weaknesses, and relevance of these works to the proposed mechanism, rather than simply listing them.

Inadequate explanation of threat modeling: The threat modeling section lacks specific details about the identified threats and their potential impact on the mechanism. Providing concrete examples and discussing potential attack vectors would enhance the validity and usefulness of the threat modeling process.

Lack of detailed methodology description: The methodology section lacks sufficient details and step-by-step instructions for each stage of the workflow. This makes it difficult for readers to replicate the process in their own research. Providing more specific explanations and examples would improve the clarity and reproducibility of the methodology.

Incomplete explanations of state machine diagrams and timed automata models: While the manuscript presents state machine diagrams and timed automata models, the descriptions and explanations of state transitions and component interactions are insufficient. Providing more detailed explanations and examples would improve the clarity and understanding of the modeling process.

Limited discussion of verification results: The manuscript lacks a comprehensive discussion of the significance and implications of the formal verification results. It is important to provide insights into the strengths and weaknesses of the mechanism based on the verification outcomes, as well as discussing any potential limitations or trade-offs.

Inadequate discussion of practical implications: The discussion section briefly touches upon potential improvements and future directions, but fails to thoroughly analyze the practical implications and limitations of the proposed mechanism. Addressing scalability issues, deployment challenges, and real-world applicability would enhance the manuscript's relevance.

Lack of broader impact discussion: The conclusion could be strengthened by explicitly discussing the broader impact of the research and its contribution to the field of secure authentication for restricted devices and LPWANs. Highlighting how the proposed mechanism addresses current industry needs and its potential for widespread adoption would add value to the manuscript.

Overall, the manuscript requires improvements in terms of clarity, depth of analysis, and the inclusion of practical implications. Addressing these shortcomings would enhance the manuscript's overall quality and impact.

English is fluent and easy to understand. 

Reviewer 3 Report

The manuscript highlights the security challenges on the Internet of Things (IoT) domain, specifically related to the authentication of constrained devices in long-distance and low-throughput networks. It emphasizes the limited exploration of verifying the security of components and their non-locking behaviour in existing research. The work aims to address this gap by analysing the design-time security of a multi-factor authentication mechanism with a reputation component that goes beyond traditional data encryption and secrecy in transmission.

 Remarks:

1.      While the abstract provides a brief overview of the research's objectives, it lacks specific details about the methodology, evaluation, and findings. The abstract mentions "great security challenges" in IoT without specifying the nature of these challenges. Providing more context on the specific security threats faced by constrained devices in long-distance and low-throughput networks would have made the abstract more informative – please address this issue within the abstract.

2.      The abstract does not mention how the proposed mechanism compares to existing authentication mechanisms or whether any benchmarking or comparison was performed. Without such comparisons, it is difficult to assess the relative strengths and weaknesses of the proposed mechanism – please address this issue within the abstract.

3.      The abstract does not discuss the generalizability of the proposed mechanism to different IoT environments or provide insights into its applicability in real-world scenarios. Understanding the mechanism's scalability, compatibility, and practicality would be valuable information – please address this issue within the abstract.

4.      In discussions the article mentions refinements to state diagrams and modifications to state machines (STMs) and timed automata (ATs) due to challenges encountered during simulation. However, it does not elaborate on the nature of these challenges, the specific modifications made, or how they improved the mechanism. Without this information, it is challenging to understand the significance of these refinements - please address this issue within the paper.

5.      In discussions the article briefly mentions the manual conversion of UML diagrams to automata and the adoption of timed automata (TA). However, it does not provide a clear rationale for these choices or explain their advantages and disadvantages compared to alternative approaches. More detailed insights into the decision-making process would have been helpful - please address this issue within the paper.

6.      In discussions the article mentions the inclusion of 20 related works that used different approaches of temporal logic. However, it does not provide a critical analysis or synthesis of these related works, such as their strengths, limitations, or how they contribute to the field. Such analysis would have helped contextualize the research and highlight its novel aspects – please address this issue within the paper.

7.      The article mentions into the conclusions that the mechanism was formally verified using ATs and TCTL, demonstrating correctness and verification of properties created from security requirements. However, it does not provide specific information on the properties verified, the security requirements addressed, or the specific results of the verification process. Without these details, it is difficult to assess the effectiveness and significance of the formal verification information – please address this issue within the paper.

8.      The article highlights into the conclusions the creation of a framework of artifacts to support the verification process and the development of similar mechanisms. However, it does not provide a clear explanation of these artifacts or how they contribute to the research. Providing more specific information about the artifacts and their reusability would enhance the understanding of their value – please address this issue within the paper.

9.      In discussions the article mentions the three stages of threat modelling but does not provide specific details on the threats identified, their categorization, or the risk assessment conducted. Without this information, it is challenging to evaluate the thoroughness and effectiveness of the threat modelling process and its contribution to the formal verification - please address this issue within the paper.

10.   The article briefly mentions the future work, including analysing additional security aspects, using tools for validation, and conducting a more in-depth threat analysis. However, it does not provide a clear rationale for these future directions or how they relate to the current research's limitations or gaps. Providing a more detailed explanation of the motivations and expected outcomes of the future work would enhance the reader's understanding – please address this issue within the paper.

11. The article does not provide a comprehensive discussion of the broader implications of the research, such as its potential impact on the field of IoT security or how the verified mechanism addresses the identified security challenges. A deeper analysis of these aspects would provide a clearer understanding of the research's significance – please address this issue within the paper.

Minor editing of English language required

Round 2

Reviewer 1 Report

Thanks for addressing my comments. The paper has been substantially improved and is now in a suitable state for publication. 

N/A

Reviewer 2 Report

The reviewer congratulates the authors for improving the manuscript by addressing the reviewer's comments. 

Please check the entire manuscript for grammar and spelling mistakes before the final submission for publication. 

Reviewer 3 Report

The paper can be published in its current form.